# MiDAS 4: A global catalogue of full-length 16S rRNA gene sequences and taxonomy for studies of bacterial communities in wastewater treatment plants

Morten Kam Dahl Dueholm [1✉], Marta Nierychlo[1], Kasper Skytte Andersen [1], Vibeke Rudkjøbing[1], Simon Knutsson[1], MiDAS Global Consortium*, Mads Albertsen [1] & Per Halkjær Nielsen [1✉]

Microbial communities are responsible for biological wastewater treatment, but our knowledge of their diversity and function is still poor. Here, we sequence more than 5 million high-quality, full-length 16S rRNA gene sequences from 740 wastewater treatment plants (WWTPs) across the world and use the sequences to construct the 'MiDAS 4' database. MiDAS 4 is an amplicon sequence variant resolved, full-length 16S rRNA gene reference database with a comprehensive taxonomy from domain to species level for all sequences. We use an independent dataset (269 WWTPs) to show that MiDAS 4, compared to commonly used universal reference databases, provides a better coverage for WWTP bacteria and an improved rate of genus and species level classification. Taking advantage of MiDAS 4, we carry out an amplicon-based, global-scale microbial community profiling of activated sludge plants using two common sets of primers targeting regions of the 16S rRNA gene, revealing how environmental conditions and biogeography shape the activated sludge microbiota. We also identify core and conditionally rare or abundant taxa, encompassing 966 genera and 1530 species that represent approximately 80% and 50% of the accumulated read abundance, respectively. Finally, we show that for well-studied functional guilds, such as nitrifiers or polyphosphate-accumulating organisms, the same genera are prevalent worldwide, with only a few abundant species in each genus.

[1] Center for Microbial Communities, Department of Chemistry and Bioscience, Aalborg University, Aalborg, Denmark. *A list of authors and their affiliations appears at the end of the paper. ✉email: md@bio.aau.dk; phn@bio.aau.dk

The activated sludge process for biological treatment of wastewater was discovered ~100 years ago[1] and is now the world's largest application of biotechnology by volume[2]. The process relies on microbial degradation of organic and inorganic compounds, biotransformation of toxic substances and removal of pathogens. However, wastewater contains many resources, which are currently poorly exploited. To meet the UN sustainable development goals, a transition is taking place to integrate treatment with the recovery of resources (phosphorus, nitrogen, clean water, etc.) and energy production (biogas)[3,4]. Activated sludge and other treatment systems, such as granular sludge and biofilters, all rely on complex microbial communities. Advances in the understanding of the microbial ecology of these communities have been ongoing for decades. However, many important microbes remain unidentified and their ecophysiology and ecology undescribed.

While several studies have attempted to resolve the microbial diversity in wastewater treatment plants (WWTPs), most have focused on few facilities in specific countries or regions[5–12]. The only comprehensive global diversity study of WWTPs concluded that there are billions of different species-level OTUs (97% identity, 16S rRNA V4 region), and that very few OTUs are shared across the world[2]. The 28 core OTUs identified in the study only accounted for 12% of accumulated read abundance in the samples, suggesting that we deal with an overwhelming microbial diversity and complexity. However, studies of process-critical functional groups have indicated that their global diversity could be much lower, especially if we focus only on the abundant species, which are likely to have a notable impact on treatment performance[13].

Nearly all microbial community studies of WWTPs are seriously hampered by two problems that limit our insight and ability to share knowledge: The lack of standardised protocols and the lack of comprehensive 16S rRNA gene reference databases. In order to address these problems, we developed the Microbial Database for Activated Sludge (MiDAS), an ecosystem-specific platform for wastewater treatment systems[14–16]. In the MiDAS project, we have thoroughly evaluated different wet-lab protocols, e.g. DNA extraction methods, and choice of amplicon primers and amplicon library preparation[17] and we provide standardised protocols online (https://www.midasfieldguide.org/guide/protocols). The second issue is the use of different 16S rRNA gene reference databases that lack reference sequences with high identity to those present in WTTPs for many microbes, and also lack a comprehensive taxonomy for the many uncultured environmental taxa[18]. To overcome these problems, we recently developed MiDAS 3, an ecosystem-specific full-length 16S rRNA gene reference database for wastewater treatment systems[16,18]. Although MiDAS 3 is only based on Danish nutrient removal plants and anaerobic digesters, it also performed well on samples from similar plants in other countries[18]. However, more plant designs, process types, and geographical locations are needed to cover the global microbial diversity in WWTPs at the highest taxonomic resolution.

Here we present a large global WWTP sampling and sequencing campaign with samples from 740 WWTPs. More than 5 million high-quality, full-length 16S rRNA gene sequences were obtained and used to expand MiDAS 3 to cover the global diversity of microbes in wastewater treatment systems. The resulting database and taxonomy (MiDAS 4) represent a comprehensive catalogue that may act as a common vocabulary for linking microbial taxonomy with function among studies across the field. Furthermore, we carried out amplicon surveys on all activated sludge samples obtained based on the two commonly applied amplicon primer sets targeting V1–V3 and V4 regions. With this data, we (i) evaluate the coverage and taxonomic resolution of the two primer sets for microbial community profiling of WWTPs, (ii) determine which environmental and geographic parameters correlate with specific genera, (iii) identify process-important taxa, and (iv) investigate the genus- and species-level diversity within important functional guilds.

## Results and discussion

The MiDAS global consortium was established in 2018 to coordinate the sampling and collection of metadata from WWTPs across the globe (Supplementary Data 1). Samples were obtained in duplicates from 740 WWTPs in 425 cities, 31 countries on six continents (Fig. 1a). The majority of the WWTPs were configured with the activated sludge process (69.7%) (Fig. 1b), and these were the main focus of the subsequent analyses. Nevertheless, WWTPs based on biofilters, moving bed bioreactors (MBBR), membrane bioreactors (MBR), and granular sludge were also sampled to cover the microbial diversity in other types of WWTPs. The activated sludge plants were designed for carbon removal only (C; 22.1%), carbon removal with nitrification (C,N; 9.5%), carbon removal with nitrification and denitrification (C,N,DN; 40.9%), and carbon removal with nitrogen removal and enhanced biological phosphorus removal, EBPR (C,N,DN,P; 21.7%) (Fig. 1c). The first type represents the simplest design whereas the latter represents the most advanced process type with varying oxic and anoxic stages or compartments.

**MiDAS 4: a global 16S rRNA gene catalogue and taxonomy for WWTPs.** Microbial community profiling at high taxonomic resolution (genus- and species-level) using 16S rRNA gene amplicon sequencing requires a reference database with high-identity reference sequences (≥99% sequence identity) for the majority of the bacteria in the samples and a complete seven-rank taxonomy (domain to species) for all reference sequences[16,18]. To create such a database for bacteria in WWTPs globally, we applied synthetic long-read full-length 16S rRNA gene sequencing[18,19] on samples from all WWTPs included in this study.

More than 5.2 million full-length 16S rRNA gene sequences were obtained after quality filtering and primer trimming. The sequences were processed with AutoTax[18] to yield 80,557 full-length 16S rRNA gene amplicon sequence variant (FL-ASVs). These reference sequences were added to our previous MiDAS 3 database[16], providing a combined database (MiDAS 4) with a total of 90,164 unique, chimera-free FL-ASV reference sequences. The absence of detectable chimeric sequences is a unique feature of the database and is achieved due to the attachment of unique molecular identifiers (UMIs) to each end of the original template molecules before any PCR amplification steps[19]. This allows filtering of true biological sequences from chimera already in the synthetic long-read assembly[18,19]. The novelty of the FL-ASVs were determined based on the percent identity shared with their closest relatives in the SILVA 138 SSURef NR99 database and the threshold for each taxonomic rank proposed by Yarza et al.[20]. Out of all FL-ASVs, 88% had relatives above the genus-level threshold (≥94.5% identity) and 56% above the species-level threshold (≥98.7% identity) (Fig. 2 and Table 1).

**MiDAS 4 provides placeholder names for many environmental taxa.** Although only a small percentage of the reference sequences in MiDAS 4 represented new putative taxa at higher ranks (phylum, class, or order) according to the sequence identity thresholds proposed by Yarza et al.[20], a large number of sequences lacked lower-rank taxonomic classifications and was assigned de novo placeholder names by AutoTax[18] (Fig. 2 and Table 1). In total, de novo taxonomic names were generated by

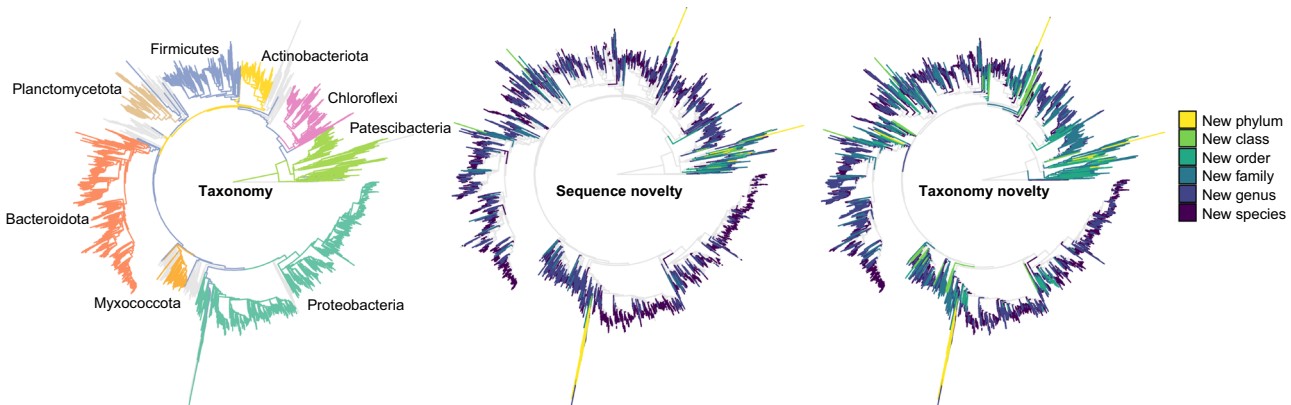

**Fig. 1 Sampling of WWTPs across the world. a** Geographical distribution of WWTPs included in the study and their process configuration. **b** Distribution of plant types. MBBR moving bed bioreactor, MBR membrane bioreactor. **c** Distribution of process types for the activated sludge plants. C carbon removal, C,N carbon removal with nitrification, C,N,DN carbon removal with nitrification and denitrification, C,N,DN,P carbon removal with nitrogen removal and enhanced biological phosphorus removal (EBPR). The values next to the bars are the number of WWTPs in each group.

**Fig. 2 Novel sequences and de novo taxa defined in the MiDAS 4 reference database.** The phylogenetic trees are based on a multiple alignment of all MiDAS 4 reference sequences, which were first aligned against the global SILVA 138 alignment using the SINA aligner, and subsequently pruned according to the ssuref:bacteria positional variability by parsimony filter in ARB to remove hypervariable regions. The eight phyla with most FL-ASVs are highlighted in different colours. Sequence novelty was determined by the percent identity between each FL-ASV and their closest relative in the SILVA_138_SSURef_Nr99 database according to Usearch mapping and the taxonomic thresholds proposed by Yarza et al.[22] shown in Table 1. Taxonomy novelty was defined based on the assignment of de novo taxa by AutoTax[20].

AutoTax for 26 phyla (30.6% of observed), 83 classes (37.2% of observed), 297 orders (46.8% of observed), and more than 8000 genera (86.3% of observed). Without the de novo taxonomy we would not be able to discuss these taxa across studies to unveil their potential role in wastewater treatment systems.

Phylum-specific phylogenetic trees were created to determine if the FL-ASV reference sequences that were assigned to de novo phyla were actual phyla or simply artifacts related to the naive sequence identity-based assignment of de novo placeholder taxonomies (Supplementary Fig. 1a). The majority (65 FL-ASVs)

**Table 1 Novel sequences and de novo taxa observed in the MiDAS 4 reference database.**

| | Sequence novelty | | Taxonomy novelty[a] | |
| --- | --- | --- | --- | --- |
| | Sequences | Percentage | De novo taxa | Percentage |
| Phylum (<75.0%) | 84 | 0.09% | 26 | 30.59% |
| Class (<78.5%) | 183 | 0.20% | 83 | 37.22% |
| Order (<82.0%) | 334 | 0.37% | 297 | 46.77% |
| Family (<86.5%) | 1,067 | 1.18% | 1,313 | 69.84% |
| Genus (<94.5%) | 10,739 | 11.91% | 8,220 | 86.33% |
| Species (<98.7%) | 40,036 | 44.40% | 30,264 | 96.54% |

Sequence novelty was determined based on the percent identity between each FL-ASV and their closest relative in the SILVA 138 SSURef NR99 database and the taxonomic thresholds proposed by Yarza et al.[22]. Taxonomy novelty was defined based on the number of de novo taxa assigned by AutoTax at each taxonomic rank.
[a]De novo species also includes known species that cannot be resolved based on full-length 16S rRNA genes.

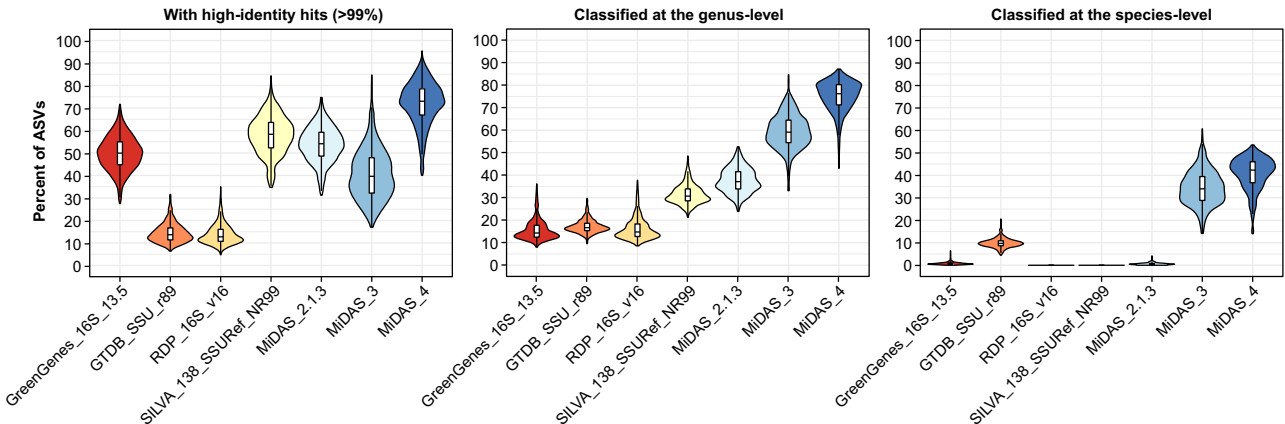

**Fig. 3 Database evaluation based on amplicon data from the Global Water Microbiome Consortium project.** Raw amplicon data from the Global Water Microbiome Consortium project[2] was processed to resolve ASVs of the 16S rRNA gene V4 region. The ASVs for each of the samples were filtered based on their relative abundance (only ASVs with ≥0.01% relative abundance were kept) before the analyses. The percentage of the microbial community represented by the remaining ASVs after the filtering was 88.35 ± 2.98% (mean ± SD) across samples. High-identity (≥99%) hits were determined by the stringent mapping of ASVs to each reference database. Classification of ASVs was done using the SINTAX classifier. The violin and box plots represent the distribution of percent of ASVs with high-identity hits or genus/species-level classifications for each database across $n = 1165$ biologically independent samples. Box plots indicate median (middle line), 25th, 75th percentile (box) and the min and max values after removing outliers based on 1.5x interquartile range (whiskers). Outliers have been removed from the box plots to ease visualisation. Different colours are used to distinguish the different databases.

created deep branches from within the Alphaproteobacteria together with 16S rRNA gene sequences from mitochondria, suggesting they represented divergent mitochondrial genes rather than true novel phyla. We also observed several FL-ASVs assigned to de novo phyla that branched from the classes Parcubacteria (3 FL-ASVs) and Microgenomatis (22 FL-ASVs) within the Patescibacteria phylum. These two classes were originally proposed as superphyla due to an unusually high rate of evolution of their 16S rRNA genes[21,22]. It is, therefore, likely that these de novo phyla are also artefacts due to the simple taxonomy assignment approach, which does not take different evolutionary rates into account[18]. Most of the class- and order-level novelty was found within the Patescibacteria, Proteobacteria, Firmicutes, Planctomycetota and Verrucomicrobiota. (Supplementary Fig. 1b). At the family- and genus-level, we also observed many de novo taxa affiliated to Bacteroidota, Bdellovibrionota and Chloroflexi.

**MiDAS 4 provides a common taxonomy for the field.** The performance of the MiDAS 4 database was evaluated based on an independent amplicon dataset from the Global Water Microbiome Consortium (GWMC) project[2], which covers ~1200 samples from 269 WWTPs. The raw GWMC amplicon data of the 16S rRNA gene V4 region was resolved into ASVs, and the percent identity to their best hits in MiDAS 4 and other reference databases was

calculated (Fig. 3). The MiDAS 4 database had high-identity hits (≥99% identity) for 72.0 ± 9.5% (mean ± SD) of GWMC ASVs with ≥0.01% relative abundance, compared to 57.9 ± 8.5% for the SILVA 138 SSURef NR99 database, which was the best of the universal reference databases (Fig. 3). The relative abundance cutoff selects taxa that likely have a quantitative impact on the ecosystem while filtering out the rare biosphere which includes many bacteria introduced with the influent wastewaters[23]. Similar analyses of ASVs obtained from the samples included in this study showed, not surprisingly, even better performance with high-identity hits for 90.7 ± 7.9% of V1–V3 ASVs and 90.0 ± 6.6% of V4 ASVs with ≥0.01% relative abundance, compared to 60.6 ± 11.9% and 73.9 ± 10.3% for SILVA (Supplementary Fig. 2a). Although the sampling of WWTPs was focused towards activated sludge plants, the MiDAS 4 database also includes high-identity references for most ASVs in other plant types (granules, biofilters, etc.) (Supplementary Fig. 2b). This suggests that most taxa were shared across plant types, although often present in other relative abundances.

Using MiDAS 4 with the SINTAX classifier, it was possible to obtain genus-level classifications for 75.0 ± 6.9% of the GWMC ASVs with ≥0.01% relative abundance (Fig. 3). In comparison, SILVA 138 SSURef NR99, which was the best of the universal reference databases, could only classify 31.4 ± 4.2% of the ASVs to genus-level. When MiDAS 4 was used to classify amplicons from

this study, we obtained genus-level classification for $92.0 \pm 4.0\%$ of V1–V3 ASVs and $84.8 \pm 3.6\%$ of V4 ASVs (Supplementary Fig. 2a). This is close to the theoretical limit set by the phylogenetic signal provided by each amplicon region analyzed[18]. Improved classifications were also observed for archaeal V4 ASVs ($93.3 \pm 10.6\%$ for MiDAS 4 vs $69.3 \pm 21.3\%$ for SILVA), although no additional archaeal reference sequences were added to the MiDAS database in this study.

MiDAS 4 was also able to assign species-level classifications to $40.8 \pm 7.1\%$ of the GWMC ASVs. In contrast, the 16S rRNA gene reference database obtained from GTDB SSU r89, which is the only universal reference database that contains a comprehensive species-level taxonomy, only classified $9.9 \pm 2.0\%$ of the ASVs (Fig. 3). For the ASVs created in this study, MiDAS 4 provided a species-level classification for $68.4 \pm 6.1\%$ of the V1–V3 and $48.5 \pm 6.0\%$ of the V4 ASVs (Supplementary Fig. 2a).

Based on the large number of WWTPs sampled, their diversity, and the independent evaluation based on the GWMC dataset[2], we expect that the MiDAS 4 reference database essentially covers the large majority of bacteria in WWTPs worldwide. Therefore, the MiDAS 4 taxonomy should act as a shared vocabulary for wastewater treatment microbiologists, providing opportunities for cross-study comparisons and ecological studies at high taxonomic resolution.

**Comparison of the V1–V3 and V4 primer sets for community profiling of WWTPs.** Before investigating what factors shape the activated sludge microbiota, we compared short-read amplicon data created for all activated sludge samples belonging to the four main process types (C; C,N; C,N,DN and C,N,DN,P) collected in the Global MiDAS project using two commonly used primer sets that target the V1–V3 or V4 variable region of the 16S rRNA gene. The V1–V3 primers were chosen because the corresponding region of the 16S rRNA gene provides the highest taxonomic resolution of common short-read amplicons[18,24], and these primers have previously shown great correspondence with metagenomic data and quantitative fluorescence in situ hybridisation (FISH) results for wastewater treatment systems[17]. The V4 region has a lower phylogenetic signal, but the primers used for amplification have better theoretical coverage of the bacterial diversity in the SILVA database[18,24].

The majority of genera (62%) showed less than twofold difference in relative abundances between the two primer sets, and the rest were preferentially detected with either the V1–V3 or the V4 primer (19% for both) (Fig. 4). We observed that several genera of known importance detected in high abundance by V1–V3 were hardly observed by V4, including *Acidovorax*, *Rhodoferax*, *Ca*. Villigracilis, *Sphaerotilus* and *Leptothrix*. Similarly, we observed genera abundant with V4 but strongly underestimated by V1–V3, such as *Acinetobacter* and *Prosthecobacter*. A complete list of differentially detected genera (Supplementary Data 2) serves as a valuable tool in combination with in silico primer evaluation for deciding which primer pair to use for targeted studies of specific taxa.

Because the V1–V3 primers provide better classification rates at the genus- and species-level (Supplementary Fig. 2a), we primarily focused on this dataset for the following analyses. It should be noted that the V1–V3 primer set performs poorly on anammox bacteria[25,26] and does not target archaea at all. To determine the importance of these groups, we estimated their relative read abundance using the V4 amplicon data. *Ca*. Brocadia and *Ca*. Anammoximicrobium were the only anammox genera detected, and the latter was never more than 0.6% abundant. *Ca*. Brocadia was observed in MBBR reactors and granular sludge in anammox reactors with relative read abundances reaching 29%,

but it was below 0.1% relative abundance in all but two of the activated sludge samples investigated. For archaea, the relative read abundance was generally low (median = 0.18%), but for a few WWTPs high (up to 11.7%), so archaea should not be neglected in these cases.

**Process and environmental factors affecting the activated sludge microbiota.** Alpha diversity analysis revealed that the rarefied (10,000 read per sample) richness and diversity in activated sludge plants were most strongly affected by process type, industrial load and continent (Supplementary Fig. 3 and Supplementary Note 1). The richness and diversity increased with the complexity of the treatment process, as found in other studies, reflecting the increased number of niches[27]. In contrast, it decreased with high industrial loads, presumably because industrial wastewater often is less complex and therefore promotes the growth of fewer specialised species[7]. The effect of continents is presumably caused by the necessary unbalanced sampling of WWTPs and confounded by the effects of plant types and industrial loads.

Distance decay relationship (DDR) analyses were used to determine the effect of geographic distance on the microbial community similarity of activated sludge plants with the four main process types (Supplementary Fig. 4 and Supplementary Note 2). We found that distance decay was only effective within shorter geographical distances (<2500 km), which suggests that the microbiota was partly shaped by immigrating bacteria from the source community as recently observed[23]. In addition, we observed low similarity between geographically separated samples (>2500 km) at the ASV-level, but higher similarities with OTUs clustered at 97% and even more at the genus-level. This suggests that many ASVs are geographically restricted and functionally redundant in the activated sludge microbiota, so different strains or species from the same genus across the world may provide similar functions.

To gain a deeper understanding of the factors that shape the activated sludge microbiota, we examined the genus-level taxonomic beta-diversity using principal coordinate analysis (PCoA) and permutational multivariate analysis of variance (PERMANOVA) analyses (Fig. 5 and Supplementary Note 3). We have chosen taxonomic diversity instead of phylogenetic diversity (UniFrac) because many of the important traits are categorical (yes/no) and only conserved at lower taxonomic ranks (genus/species). The analysis was made at the genus-level due to the high classification rate achieved with MiDAS 4 and because genera were less affected by DDR compared to ASVs. We found that the overall microbial community was most strongly affected by continent and temperature in the WWTPs. However, process type, industrial load and the climate zone also had significant impacts. The percentage of total variation explained by each parameter was generally low, indicating that the global WWTPs microbiota represents a continuous distribution rather than distinct states, as observed for the human gut microbiota[28].

**Genera selected for by process type and temperature.** Redundancy analyses (RDA) were used to identify which genera were the strongest indicators for specific processes and/or environmental conditions. RDA was performed on both V1–V3 (Supplementary Fig. 5) and V4 (Supplementary Fig. 6) amplicon datasets to ensure that essential taxa were not missed due to primer bias. We here highlight the results for process type and temperature. Results for the other parameters and RDA scores for all analyses can be found in Supplementary Note 4 and Supplementary Data 3, respectively.

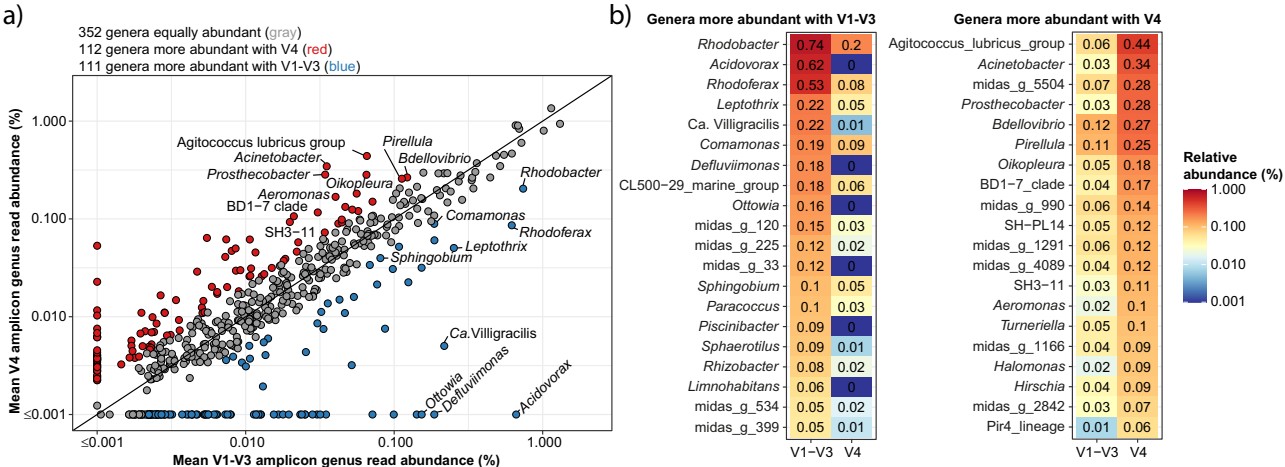

**Fig. 4 Comparison of relative genus abundance based on V1–V3 and V4 region 16S rRNA gene amplicon data. a** Mean relative abundance was calculated based on 709 activated sludge samples. Genera present at ≥0.001% relative abundance in V1–V3 and/or V4 datasets are considered. Genera with less than twofold difference in relative abundance between the two primer sets are shown with gray circles, and those that are overrepresented by at least twofold with one of the primer sets are shown in red (V4) and blue (V1–V3). The twofold difference is an arbitrary choice; however, it relates to the uncertainty we usually encounter in amplicon data. Genus names are shown for all taxa present at a minimum of 0.1% mean relative abundance (excluding those with de novo names). **b** Heatmaps of the most abundant genera with more than twofold relative abundance difference between the two primer sets.

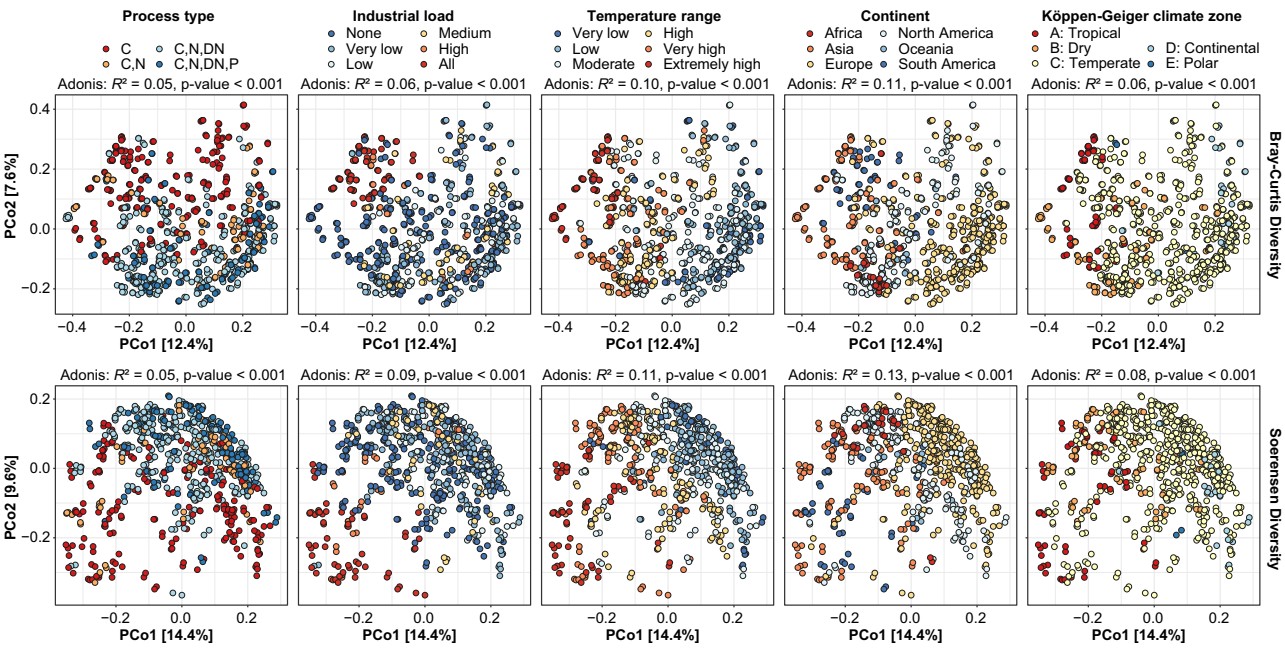

**Fig. 5 Effects of process and environmental factors on the activated sludge microbial community structure. Principal coordinate analyses of Bray–Curtis and Soerensen beta-diversity for genera based on V1–V3 amplicon data. Samples are coloured based on metadata.** The fraction of variation in the microbial community explained by each variable in isolation was determined by PERMANOVA (Adonis $R^2$-values). Exact $P$ values <0.001 could not be confidently determined due to the permutational nature of the test. Process types: C carbon removal, C,N carbon removal with nitrification, C,N,DN carbon removal with nitrification and denitrification, C,N,DN,P carbon removal with nitrogen removal and enhanced biological phosphorus removal (EBPR). Temperature range: Very low = <10 °C, low = 10–15 °C, moderate = 15–20 °C, high = 20–25 °C, very high = 25–30 °C, extremely high = >30 °C. Industrial load: None = 0%, very low = 0–10%, low = 10–30%, medium = 30–50%, high 50–100%, all = 100%.

The RDA analyses of process types revealed that genera commonly involved in nitrification (*Nitrosomonas* and *Nitrospira*), denitrification (*Rhodoferax*, *Sulfuritalea*) and the polyphosphate-accumulating organisms (PAOs) (*Tetrasphaera*, *Ca.* Accumulibacter and *Dechloromonas*) were strongly enriched in more advanced process types along with de novo taxa midas_g_17 (family: Saprospiraceae), midas_g_72 (class: Anaerolineae) and midas_g_300 (order: Sphingobacteriales). Conversely, carbon removal plants were enriched with *Hydrogenophaga*

and *Prevotella*, the filamentous genera *Sphaerotilus* and *Thiothrix*, and the glycogen-accumulating organisms (GAOs) *Ca.* Competibacter and *Defluviicoccus*. Specific to the EBPR plants were an increased abundance of known PAOs (see above) and *Azospira*, *Propionivibrio*, *Propioniciclava*, *Ca.* Amarolinea and the de novo taxa midas_g_399 (class: Actinobacteria), midas_g_384 (family: Saprospiraceae) and midas_g_945 (class: Elusimicrobia). The latter genera should be considered targets for further characterisation as potential PAOs or GAOs.

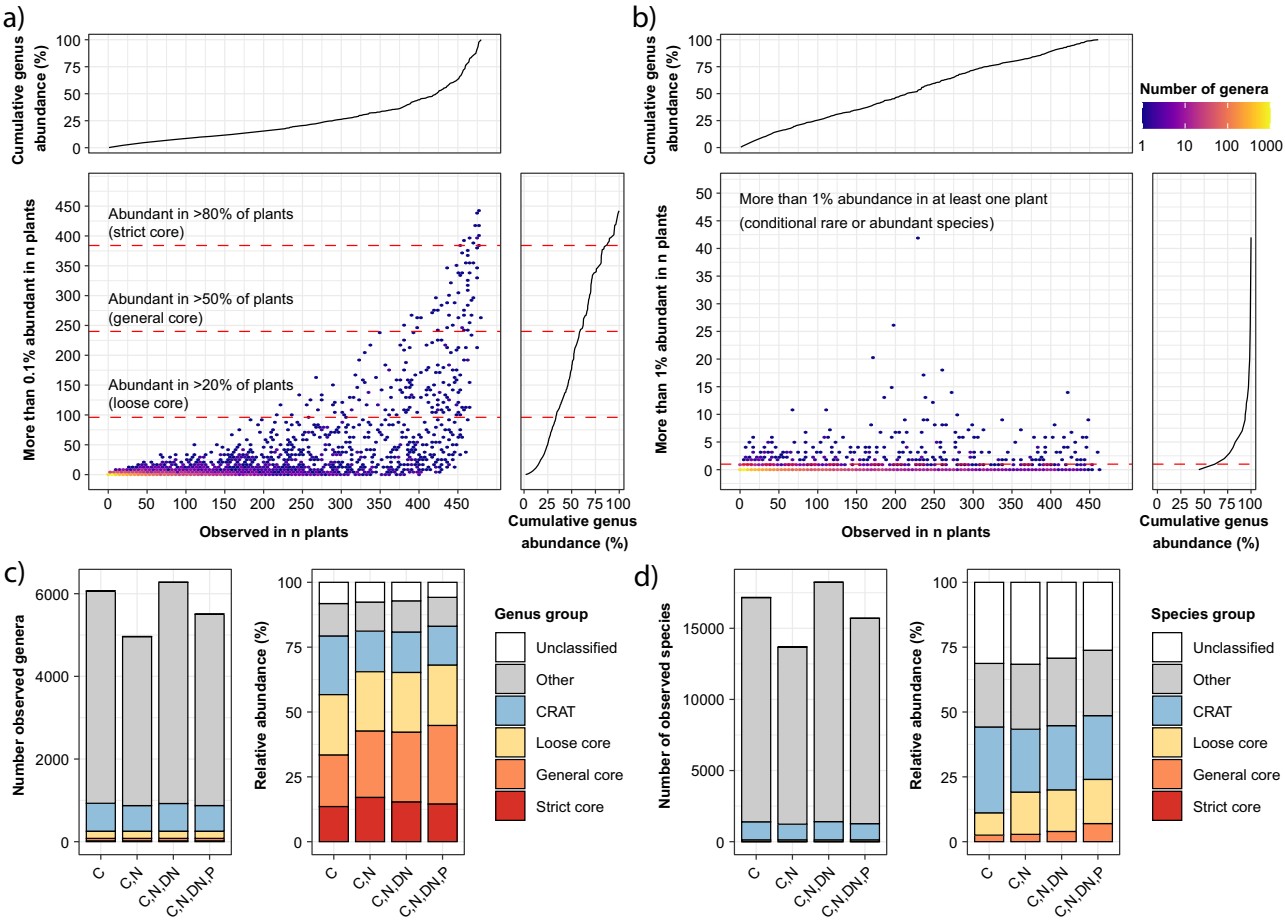

**Fig. 6 Identification of core and conditionally rare or abundant taxa based on V1–V3 amplicon data. a** Identification of strict, general and loose core genera based on how often a given genus was observed at a relative abundance above 0.1% in WWTPs. **b** Identification of conditionally rare or abundant (CRAT) genera based on whether a given genus was observed at a relative abundance above 1% in at least one WWTP. The cumulative genus abundance is based on all ASVs classified at the genus-level. All core genera were removed before identification of the CRAT genera. **c**, **d** Number of genera and species, respectively, and their abundance in different process types across the global WWTPs. Values for genera and species are divided into strict core, general core, loose core, CRAT, other taxa and unclassified ASVs. The relative abundance of different groups was calculated based on the mean relative abundance of individual genera or species across samples. C carbon removal, C,N carbon removal with nitrification, C,N,DN carbon removal with nitrification and denitrification, C,N,DN,P carbon removal with nitrogen removal and enhanced biological phosphorus removal (EBPR).

The RDA based on temperature showed that high temperatures were associated with an increased abundance of *Ca.* Competibacter, *Thauera*, *Defluviicoccus*, *Azospira*, *Rhodoplanes*, *Ottowia* and *Phaeodactylibacter*, whereas lower temperatures favoured the presence of *Flavobacterium*, *Tetrasphaera*, *Ferruginibacter*, *Trichococcus*, *Ca.* Epiflobacter and *Acinetobacter*. These differences suggest that plants with similar designs and operations may have differences in community structure depending on prevailing temperature conditions.

**Core and conditional rare or abundant taxa in the global activated sludge microbiota.** Core taxa are commonly defined in complex communities based on how frequently specific taxa are observed in samples from a well-defined habitat[29]. In addition, an abundance threshold can be applied to select for those taxa that may likely have a quantitative impact on ecosystem functioning[6]. We here used three frequency thresholds for the core taxa with >0.1% relative abundance in 80% (strict core), 50% (general core) and 20% (loose core) of all activated sludge plants (Fig. 6a).

In addition to the core taxa, we also identified conditionally rare or abundant taxa (CRAT)[30] (Fig. 6b). These are taxa typically present in low abundance but occasionally become prevalent, including taxa related to process disturbances, such as bacteria

causing activated sludge foaming or those associated with the degradation of specific residues in industrial wastewater. CRAT have only been studied in a single WWTP treating brewery wastewater, despite their potential effect on performance[30,31]. CRAT are here defined as taxa which are not part of the core, but present in at least one WWTP with a relative abundance above 1%.

Core taxa and CRAT were identified for both the V1–V3 and V4 amplicon data to ensure that critical taxa were not missed due to primer bias. We identified 250 core genera (15 strict, 65 general and 170 loose) and 715 CRAT genera (Supplementary Data 4). The strict core genera (Fig. 7) mainly contained genera with versatile metabolisms found in several environments, including *Flavobacterium*, *Novosphingobium* and *Haliangium*. The general core (Fig. 7) included many known bacteria associated with nitrification (*Nitrosomonas* and *Nitrospira*), polyphosphate accumulation (*Tetrasphaera*, *Ca.* Accumulibacter) and glycogen accumulation (*Ca.* Competibacter). The loose core contained well-known filamentous bacteria (*Ca.* Microthrix, *Ca.* Promineofilum, *Ca.* Sarcinithrix, *Gordonia*, *Kouleothrix* and *Thiothrix*), but also *Nitrotoga*, a less common nitrifier in WWTPs.

Because MiDAS 4 allowed for species-level classification, we also identified core and CRAT species based on the same criteria

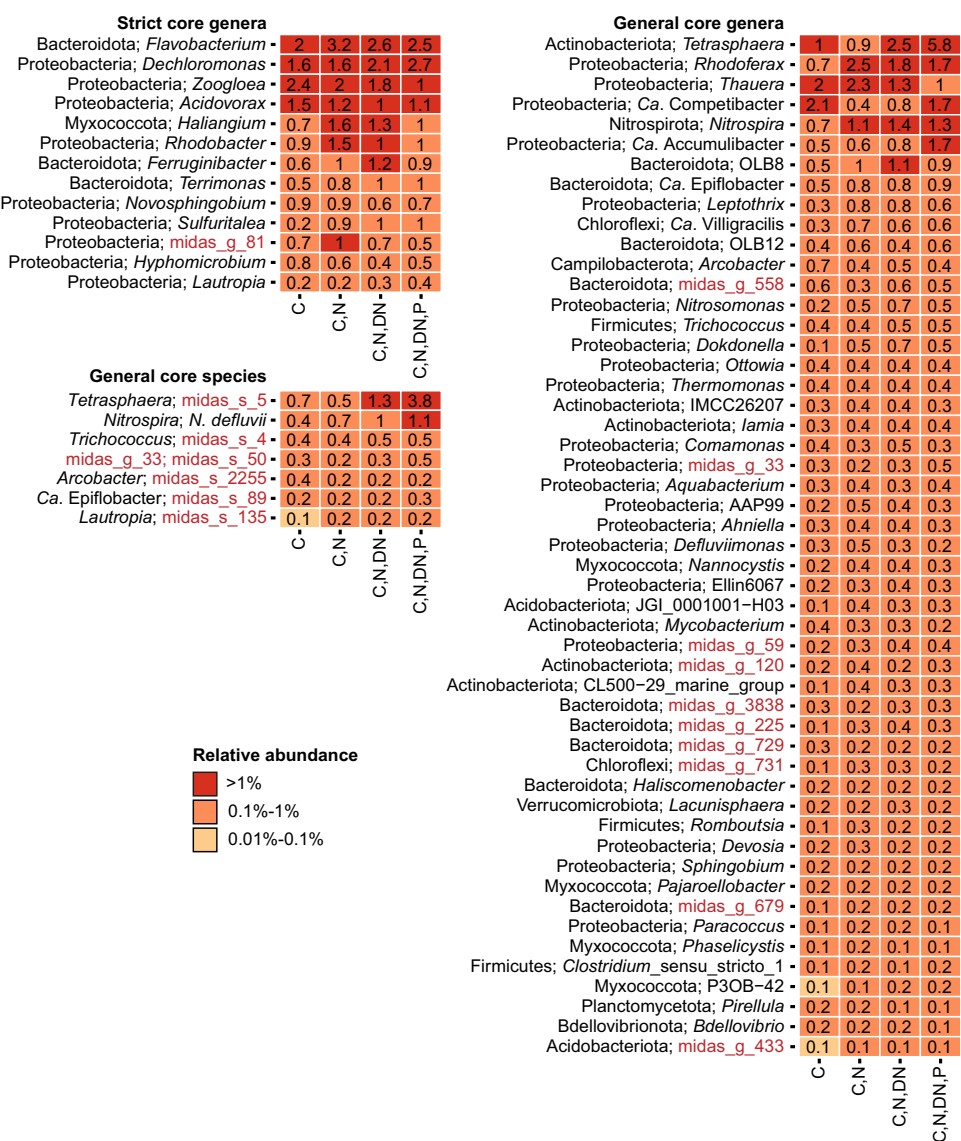

**Fig. 7 Percent relative abundance of strict and general core taxa across process types.** The taxonomy for the core genera indicates phylum and genus. For general core species, genus names are also provided. De novo taxa in the core are highlighted in red. C carbon removal, C,N carbon removal with nitrification, C,N,DN carbon removal with nitrification and denitrification, C,N,DN,P carbon removal with nitrogen removal and enhanced biological phosphorus removal (EBPR).

as for genera (Supplementary Fig. 7 and Supplementary Data 4). This revealed 113 core species (0 strict, 9 general and 104 loose). The general core species (Fig. 7) included *Nitrospira defluvii* and *Tetrasphaera* midas_s_5, a common nitrifier and PAO, respectively. *Arcobacter* midas_s_2255, a potential pathogen commonly abundant in the influent wastewater, was also part of the general core[32]. The loose core contained additional species associated with nitrification (*Nitrosomonas* midas_s_139 and *Nitrospira nitrosa*), polyphosphate accumulation (*Ca.* Accumulibacter phosphatis, *Dechloromonas* midas_s_173, *Tetrasphaera* midas_s_45), as well as known filamentous species (*Ca.* Microthrix parvicella and midas_s_2 (recently named *Ca.* M. subdominans[33]), *Ca.* Villigracilis midas_s_471 and midas_s_9223, *Leptothrix* midas_s_884). In addition to the core species, we identified 1417 CRAT species. As CRAT taxa are generally found in low abundance and the current study does not include time series or influent data, we cannot say anything conclusive about their general implications for the ecosystem. However, they may be present due to short-term mass

immigration[23] or specific operational conditions[34] and in both cases, potentially affect the plant operation. They should therefore be considered important target for further investigations together with the core taxa.

**Many core taxa and CRAT can only be identified with MiDAS 4.** The core taxa and CRAT included a large proportion of MiDAS 4 de novo taxa. At the genus-level, 106/250 (42%) of the core genera and 500/715 (70%) of the CRAT genera had MiDAS placeholder names. At the species-level, the proportion was even higher. Here placeholder names were assigned to 101/113 (89%) of the core species and 1352/1417 (95%) CRAT species. This highlights the importance of a comprehensive taxonomy that includes the uncultured environmental taxa.

**The core and CRAT taxa cover the majority of the global activated sludge microbiota.** Although the core taxa and CRAT represent a small fraction of the total diversity observed in the

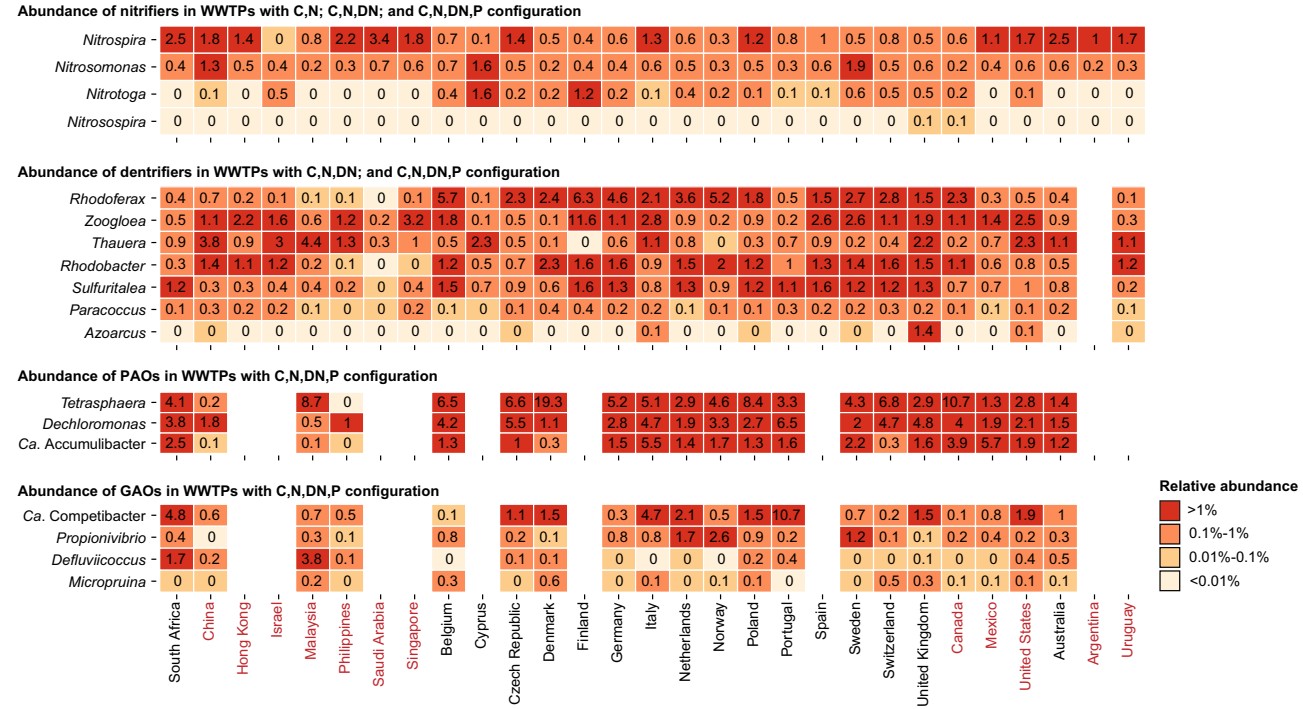

**Fig. 8 Global diversity of genera belonging to major functional groups.** The percent relative abundance represents the mean abundance for each country considering only WWTPs with the relevant process types. Countries are grouped based on continent (shifting colour).

MiDAS 4 reference database, they accounted for the majority of the observed global activated sludge microbiota (Fig. 6c, d). Accumulated read abundance estimates ranged from 57–68% for the core genera and 11–13% for the CRAT, and combined they accounted for 68–79% of total read abundance in the WWTPs depending on process types. The core taxa represented a larger proportion of the activated sludge microbiota for the more advanced process types, which likely reflects the requirement of more versatile bacteria associated with the alternating redox conditions in these types of WWTPs. The remaining fraction, 21–32%, consisted of 6–8% unclassified genera and genera present in very low abundance, presumably with minor importance for the plant performance. The species-level core taxa and CRAT represented 11–24% and 24–33% accumulated read abundance, respectively. Combined, they accounted for almost 50% of the observed microbiota.

**Global diversity within important functional guilds.** The general change from simple to advanced WWTPs with nutrient removal and the transition to water resource recovery facilities (WRRFs) requires increased knowledge about the bacteria responsible for the removal and recovery of nutrients, so we examined the global diversity of well-described nitrifiers, denitrifiers, PAOs and GAOs (Fig. 8). GAOs were included because they may compete with the PAOs for nutrients and thereby interfere with the biological recovery of phosphorus[35]. Because MiDAS 4 provided species-level resolution for a large proportion of activated sludge microbiota, we also investigated the species-level diversity within genera affiliated with the functional guilds. A complete overview of species in all genera detected in this global study is provided in the MiDAS field guide (https://www.midasfieldguide.org/guide).

*Nitrosomonas* and potential comammox *Nitrospira* were the only abundant (≥0.1% average relative abundance) genera found among ammonia-oxidising bacteria (AOBs), whereas both *Nitrospira* and *Nitrotoga* were abundant among the nitrite

oxidisers (NOBs), with *Nitrospira* being the most abundant across all countries (Fig. 8). *Nitrobacter* was not detected, and *Nitrosospira* was detected in only a few plants in very low abundance (≤0.01% average relative abundance). At the species-level, each genus had 2–5 abundant species (Supplementary Fig. 8). The most abundant and widespread *Nitrosomonas* species was midas_s_139. However, midas_s_11707 and midas_s_11733 were dominating in a few countries. For *Nitrospira*, the most abundant species in nearly all countries was *N. defluvii*. ASVs classified as the comammox *N. nitrosa*[36,37] was also common in many countries across the world. However, because the comammox trait is not phylogenetically conserved at the 16S rRNA gene level[36,37], we cannot conclude that these ASVs represent true comammox bacteria. For *Nitrotoga*, only two species were detected with notable abundance, midas_s_181 and midas_s_9575. Ammonia-oxidising archaea (AOAs) were not detected with MiDAS 4 due to the lack of reference sequences, and because AOAs are not targeted by the V1–V3 primer pair. However, analyses of our V4 amplicon dataset classified with the SILVA database revealed a considerable relative read abundance of AOAs in Malaysia and the Philippines, but absence or low abundance of AOAs in other countries (Supplementary Fig. 9). Other studies have occasionally found AOAs across the world, but generally in lower abundance than AOBs[38–40]. To ensure detection of AOAs with MiDAS 4, we anticipate adding external reference sequences for AOAs in a future release of the database.

Denitrifying bacteria are very common in advanced activated sludge plants, but are generally poorly described. Among the known genera, *Rhodoferax*, *Zoogloea* and *Thauera* were most abundant (Fig. 8). *Zoogloea* and *Thauera* are well-known floc formers, sometimes causing unwanted slime formation[41]. *Rhodoferax* was the most common denitrifier in Europe, whereas *Thauera* dominated in Asia. Many denitrifiers could not be classified at the species-level (Supplementary Fig. 10), likely due to highly conserved 16S rRNA genes. An exception was *Zoogloea*, where midas_s_1080 and *Z. caeni* and were the most abundant species worldwide.

EBPR is performed by PAOs, with three genera recognised as important in full-scale WWTPs: *Tetrasphaera*, *Dechloromonas* and *Ca.* Accumulibacter[13]. According to relative read abundance, all three were found in EBPR plants globally, with *Tetrasphaera* as the most prevalent (Fig. 8). *Dechloromonas* was also abundant in nitrifying and denitrifying plants without EBPR, indicating a more diverse ecology. Four recognised GAOs were found globally: *Ca.* Competibacter, *Defluviicoccus*, *Propionivibrio* and *Micropruina*, with *Ca.* Competibacter being the most abundant (Fig. 8). Only a few species (2–6 species) in each genus were dominant across the world for both PAOs (Supplementary Fig. 11) and GAOs (Supplementary Fig. 12), except for *Ca.* Competibacter, which covered ~20 abundant but country-specific species. Among PAOs, the abundant species were *Tetrasphaera* midas_s_5, *Dechloromonas* midas_s_173, (recently named D. phosphorivorans) *Ca.* Accumulibacter midas_s_315, *Ca.* A. phosphatis and *Ca.* A. aalborgensis. Interestingly, some of the most abundant PAOs and GAOs were also abundant in the simple process design with C-removal, indicating more versatile metabolisms.

**Global diversity of filamentous bacteria**. Filamentous bacteria are essential for creating strong activated sludge flocs. However, in large numbers, they can also lead to loose flocs and poor settling properties. This is known as bulking, a major operational problem in many WWTPs. Many can also form foam on top of process tanks due to hydrophobic surfaces. Presently, approximately 20 genera are known to contain filamentous species[42], and among those, the most abundant are *Ca.* Microthrix, *Leptothrix*, *Ca.* Villigracilis, *Trichococcus* and *Sphaerotilus* (Fig. 9). They are all well-known from studies on mitigation of poor settling properties in WWTPs. Interestingly, *Leptothrix*, *Sphaerotilus* and *Ca.* Villigracilis belong to the genera where abundance-estimation depended strongly on primers, with V4 underestimating their abundance (Fig. 3). *Ca.* Microthrix and *Leptothrix* were strongly associated with continents, most common in Europe and less in Asia and North America (Fig. 9).

Many of the filamentous bacteria were linked to specific process types (Supplementary Fig. 13), e.g. *Ca.* Microthrix were not observed in WWTPs with carbon removal only, and *Ca.* Amarolinea were only abundant in plants with nutrient removal. The number of abundant species within the genera were generally low, with one species in *Trichococcus*, two in *Ca.* Microthrix and approximately five in *Leptothrix* and *Ca.* Villigracilis (Supplementary Fig. 14). Only five abundant species were observed for *Sphaerotilus*. However, a substantial fraction of unclassified ASVs

was also observed, demonstrating that certain species within this genus are poorly resolved based on the 16S rRNA gene. *Ca.* Promineofilum was also poorly resolved at the species-level (Supplementary Fig. 15).

**Conclusion and perspectives**. We present a worldwide collaborative effort to produce MiDAS 4, an ASV-resolved full-length 16S rRNA gene reference database, which covers more than 31,000 species and enables genus- to species-level resolution in microbial community profiling studies. MiDAS 4 covers the vast majority of WWTP bacteria globally and provides a strongly needed common taxonomy for the field, which provides the foundation for comprehensive linking of microbial taxa in the ecosystem with their functional traits. Presently, hundreds of studies are undertaken to combine engineering and microbial aspects of full-scale WWTPs. However, most ASVs or OTUs in these studies are classified at poor taxonomic resolution (family-level or above) due to the use of incomplete universal reference databases. Because many important functional traits are only conserved at high taxonomic resolution (genus- or species-level), this strongly hampers our ability to transfer taxa-specific knowledge from one study to another. This will change with MiDAS 4, and we expect that reprocessing of data from earlier studies may reveal new perspectives into wastewater treatment microbiology. Our online Global MiDAS Field Guide presents the data generated in this study and summarises present knowledge about all taxa. We encourage researchers within the field to contribute new knowledge to MiDAS using the contact link in the MiDAS website (https://www.midasfieldguide.org/guide/contact).

The global microbiota of activated sludge plants has been predicted to harbour a massive diversity with up to one billion species[2]. However, most of these occur at very low abundance and are of little importance for the treatment process. By focusing only on the abundant taxa, we can see that this number is much smaller, i.e., ~1000 genera and 1500 species. We consider these taxa functionally the most important globally, representing a "most wanted list" for future studies. Some taxa are abundant in most WWTPs (core taxa), and others are occasionally abundant in fewer plants (CRAT). The CRAT have received little attention in the field of wastewater treatment, but they can be of profound importance for WWTP performance. Both groups have a high fraction of poorly characterised species. The high taxonomic resolution provided by MiDAS 4 enables us to identify samples where these important core taxa occur in high abundance. This provides an ideal starting point for obtaining high-quality

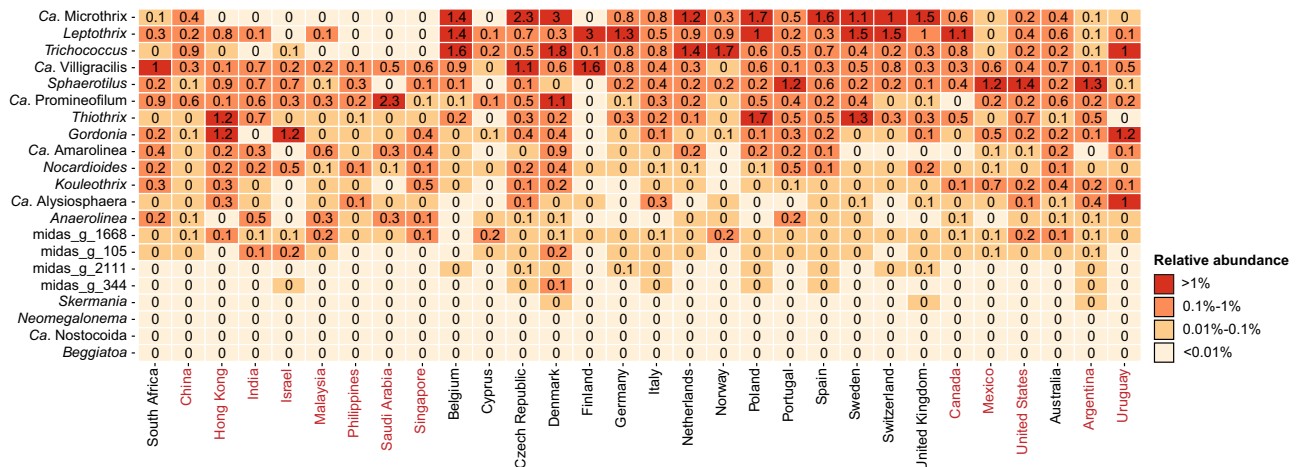

**Fig. 9 Global diversity of known filamentous organisms.** The percent relative abundance represents the mean abundance for each country across all process types. Countries are grouped based on the continent (shifting colour).

metagenome-assembled genomes (MAGs), isolation of pure cultures, in addition to targeted culture-independent studies to uncover their physiological and ecological roles.

Among the known functional guilds, such as nitrifiers or polyphosphate-accumulating organisms, the same genera were found worldwide, with only a few abundant species in each genus. There were differences in the community structure, and the abundance of dominant species was mainly shaped by process type, temperature, and in some cases, continent. This discovery sends an important message to the field: relatively few species are abundant worldwide, so research or operational results can reliably be transferred from one geographical region to another, stimulating the transition from WWTPs to more sustainable WRRFs.

The relatively low number of uncharacterised abundant species also shows that it is within our reach to describe them all in terms of identity, physiology, ecology and dynamics, providing the necessary knowledge for informed process optimisation and management. The number of poorly described genera (i.e. those with only a MiDAS placeholder genus name) was 88 among the 250 core genera (35%) and more than 89% at the species-level, so there is still some work to do to link their identities and function. An important step in this direction is the visualisation of the populations. With the comprehensive set of FL-ASVs, it is possible to design highly specific FISH probes, and to critically evaluate the old probes. In the Danish WWTPs, we have successfully done this for groups in the Acidobacteriota[43] based on the MiDAS 3 database[16]. Our recent retrieval of more than 1000 high-quality MAGs from Danish WWTPs with advanced process design is also an important step to link identity to function[44]. The HQ-MAGs can be linked directly to MiDAS 4 as they contain complete 16S rRNA genes. They cover 62% (156/250) of the core genera and 61% (69/113) of the core species identified in this study. These MAGs may also form the basis for further studies to link identity and function, e.g. by applying metatranscriptomics[45] and other in situ techniques such as FISH combined with Raman[46,47], guided by the "most wanted" list provided in this study. We expect that MiDAS 4 will have significant implications for future microbial ecology studies in wastewater treatment systems.

## Methods

**Sampling and metadata collection.** To facilitate sampling of WWTPs across the globe, we established the MiDAS global consortium, which consists of 39 wastewater treatment experts in 31 countries. Members of the consortium acted as national sampling coordinators and were in direct contact with the WWTPs. Two biological sample replicates were obtained from activated sludge aeration tanks or biomass was scraped off biofilters from each WWTP and shipped on ice to the sampling coordinators. For each replicate, 2 mL were preserved in 2 mL RNAlater (Invitrogen), stored at 4 °C until all national samples were collected (usually within a few days), and then shipped to Aalborg University with cooling elements. Upon arrival, the samples were separated into aliquots that were prepared for nucleic acid purification. Metadata associated with each WWTP was also obtained by the sampling coordinators and is provided as Supplementary Data 1. Minimum information from all plants included continent, country, GPS coordinates, sampling date, the temperature in process tank, wastewater composition (municipal vs. industrial COD fraction), process type and plant type.

**General molecular methods.** The concentration and quality of nucleic acids were determined using a Qubit 3.0 fluorometer (Thermo Fisher Scientific) and an Agilent 2200 Tapestation (Agilent Technologies), respectively. Agencourt AMPure XP beads were used as described by the manufacturer, except for the washing steps, where 80% ethanol was used. All commercial kits were used according to the protocols provided by the manufacturer unless otherwise stated.

**Nucleic acid purification.** DNA was purified using a custom plate-based extraction protocol based on the FastDNA spin kit for soil (MP Biomedicals). The protocol is available at www.midasfieldguide.org (aau_wwtp_dna_v.8.0). 320 µL of RNAlater preserved sample was pelleted by centrifugation (10,000×g, 1 min), and the pellet was resuspended in 320 µL PBS and transferred to Lysing Matrix E barcoded tubes

(MP Biomedicals). About 40 µL MT buffer was added and lysis was performed by bead beating in a FastPrep-96 bead beater (MP Biomedicals) (3 × 120 s, 1800 rpm, 2 min incubation on ice between beating). The samples were centrifuged (maximum speed, 10 min) and 200 µL supernatant was transferred to a 96-well PCR-plate. 50 µL Protein Precipitation Solution (PPS) was mixed with each sample, which was then centrifuged again. About 150 µL supernatant was cleaned-up using 100 µL Agencourt AMPure XP beads with elution into 60 µL of nuclease-free water. About 40 µL of the purified DNA was transferred to a new 96-well plate and stored at −80 °C.

**Full-length 16S rRNA gene library preparation and sequencing.** Full-length 16S rRNA gene sequencing was carried out using an improved version of our method for synthetic long-read sequencing[19]. Oligonucleotides used can be found in Supplementary Table 1. To improve amplification of bacterial 16S rRNA gene diversity for the long-read sequencing, we replaced the 1492r primer previously used[18] with the 1391r primer[48] as it has better coverage of the known bacterial diversity. In addition, we replaced the Taq polymerase used for primary amplification of the unique molecular identifier (UMI) tagged templates with a proof-reading polymerase (Supplementary Fig. 16). This increased the percentage of error-free reads nearly four-fold, which greatly improves our ability to resolve full-length 16S rRNA ASVs (FL-ASVs) from low abundant taxa because at least two identical sequences are required to resolve an FL-ASV[18].

*Adaptor annealing by PCR.* Adaptors containing barcodes and UMI and defined primer binding sites were added to each end of the bacterial 16S rRNA genes by PCR. The reaction contained 10 µL of 10x PCR Buffer (Qiagen), 2 µL of 10 mM dNTP (Qiagen), 5 µL of 10 µM f16S_pcr1_fw, 5 µL of 10 µM f16S_pcr1_rv, 4 µL of 25 mM MgCl₂, 0.5 µL of 5 U/µL Taq polymerase (Qiagen), 100 ng of pooled template DNA (from two to five WWTPs) and nuclease-free water to 100 µL. The reaction was incubated with an initial denaturation at 94 °C for 3 min followed by two cycles of denaturation at 94 °C for 30 s, annealing at 56 °C for 30 s, and extension at 72 °C for 3 min, and then a final extension at 72 °C for 5 min. The sample was purified using 0.6x AMPure XP beads and eluted in 10 µL nuclease-free water.

*Primary library amplification.* The tagged 16S rRNA gene amplicons were amplified using PCR to obtain enough product for quantification and sequencing. The reaction contained 9 µL of adaptor annealed sample, 20 µL 5× Phusion HF buffer (NEB), 2 µL of 10 mM dNTP, 5 µL of 10 µM f16S_pcr2_fw, 5 µL of 10 µM f16S_pcr2_rv, 4 µL of 25 mM MgCl, 54 µL nuclease-free water and 1 µL 2U/µL Phusion HF DNA polymerase (NEB). The reaction was incubated with an initial denaturation at 98 °C for 30 s followed by 15 cycles of denaturation at 98 °C for 10 s, annealing at 62 °C for 30 s and extension at 72 °C for 1 min and then a final extension at 72 °C for 5 min. The PCR product was purified using 0.6× AMPure XP beads and eluted in 11 µL nuclease-free water. The amplicons were validated on a D5000 screentape and quantified with the Qubit dsDNA HS Assay Kit. Up to ten samples with different barcodes were pooled with an equal amount of DNA from each sample.

*Clonal library amplification.* Tagged amplicon libraries were diluted to ~100,000 molecules/µL and amplified by PCR to obtain clonal copies of each uniquely tagged amplicon molecule. The PCR reaction contained 63.5 µL nuclease-free water, 10 µL 10X PCR buffer (Qiagen), 2 µL 10 mM dNTP, 5 µL 10 µM f16S_pcr2_fw, 5 µL 10 µM f16S_pcr2_rv, 4 µL 25 mM MgCl, 0.5 µL 5 U/µL Taq polymerase (Qiagen) and 10 µL diluted tagged amplicon product. The reaction was initiated by denaturation at 94 °C for 3 min, followed by 20 cycles of denaturation at 94 °C for 30 s, annealing at 62 °C for 30 s and extension at 72 °C for 2 min and finishing with a final extension at 72 °C for 5 min. The PCR product was purified using 0.6× AMPure XP beads with elution into 21 µL nuclease-free water. The product quality and concentration were analysed on a D5000 screentape and with the Qubit dsDNA HS Assay Kit, respectively.

*Read-tag library preparation.* A Nextera library preparation kit (Illumina) was used to prepare a paired-end read-tag sequencing library from the clonal library using a customised protocol. A tagmentation reaction was prepared with 100 ng of the clonal library in 22.5 µL nuclease-free water, 25 µL tagment DNA buffer (Illumina) and 2.5 µL tagment DNA enzyme (Illumina). The reaction was incubated at 55 °C for 5 min. The product was immediately diluted to 100 µL and purified using 0.6x AMPure XP beads with elution into 42 µL nuclease-free water.

The tagmentation products were PCR amplified using two separate PCRs (A and B). PCR A selectively amplified fragments containing the 5′ termini of the amplicons and PCR B selectively amplified fragments containing the 3′ termini. The reactions contained 20 µL purified tagmentation product, 5 µL N504 nextera adaptor (Illumina), 5 µL 10 µM f16S_readtag_fw (PCR A) or f16S_readtag_rv (PCR B) adaptor, 5 µL PCR primer cocktail (Illumina), 10 µL 5× Phusion HF buffer (NEB), 1 µL 10 mM dNTP, 3.5 µL nuclease-free water and 0.5 µL 2U/µL Phusion HF DNA polymerase (NEB). The following PCR programme was used: Initial elongation at 72 °C for 3 min, initial denaturation at 98 °C for 30 s, and 10 cycles of denaturation at 98 °C for 10 s, annealing at 60 °C for 30 s and elongation at 72 °C

for 3 min and finishing with a final extension at 72 °C for 5 min. The raw read-tag libraries were purified using 1.0x AMPure XP beads with elution into 21 μL nuclease-free water.

To ensure even sequencing coverage across the length of the 16S rRNA gene amplicons, the read-tag libraries were normalised[19]. The libraries were size-fractionated on an E-Gel CloneWell gel (Thermo Fisher Scientific). A total of 500 ng GeneRuler 1 kb DNA ladder (Thermo Fisher Scientific) was used as a reference. The gel was run until the 500 bp marker was 1 mm from the elution well, after which 20 μL elution aliquots were sampled and replaced by nuclease-free water every 15 s, up to a total of 32 aliquots. Every two aliquots were pooled, yielding 16 pooled aliquots per sample. These were then analysed on an Agilent 2200 Tapestation using the High Sensitivity D1000 Screentape. Fractions with a mean fragment length of 500–1250 bp were used for the pooling. The effective sequencing concentration for fractions from 500–950 bp was determined based on the tapestation data and the empirical formula:

$$C_{seq} = \text{Peak molarity} \times 0.0124 \times (215 - \text{Peak size}) + 10.332,$$

where the peak molarity is provided in pmol/l and the peak size in bp[19].

These fractions were pooled in equimolar concentrations ($C_{seq}$). For fractions between 950–1250 bp the entire aliquot was used for pooling (40–50 μL). The pooled aliquots were then purified using 1.0x AMPure XP beads with elution into 11 μL nuclease-free water. The quality and concentration of the coverage normalised read-tag libraries were analysed on D1000 screentapes and with the Qubit dsDNA HS Assay Kit, respectively.

*Linked-tag library preparation*. Clonal libraries were end-repaired in a reaction containing 20 ng clonal library, 2.5 μL 10× NEBNext End-Repair Reaction Buffer (New England Biolabs), 1.25 μL NEBNext End-Repair Enzyme Mix (New England Biolabs) and nuclease-free water to 25 μL. The reaction was incubated at 20 °C for 30 min. The end-repair reaction was purified using 1.0x AMPure XP beads and eluted into 10 μL nuclease-free water.

The end-repaired sample was circularised in an intramolecular blunt end ligation reaction containing 150 μL nuclease-free water, 20 μL 50% (w/w) PEG 4000 solution (Thermo Fisher Scientific), 20 μL 10× T4 DNA ligase buffer (NEB), 8 μL T4 DNA ligase (NEB), and 2 μL of the end-repaired clonal library. The reaction was incubated at 16 °C for 60 min. The circularised products were purified using 1.0× AMPure XP beads and eluted in 10 μL nuclease-free water.

The junction sequence, which contains both UMIs, were amplified by PCR in a reaction containing 8 μL of circularised clonal library, 5 μL 10× PCR buffer (Qiagen), 1 μL 10 mM dNTP mix, 2.5 with 10 μM of each f16S_linktag_fw and f16S_linktag_rv, 2 μL 25 mM MgCl₂, 30.25 μL nuclease-free water, and 0.25 μL 5 u/μL Taq polymerase (Qiagen). The PCR reaction was initiated by denaturation at 94 °C for 3 min, followed by 20 cycles of denaturation at 94 °C for 20 s, annealing at 56 °C for 20 s, and extension at 72 °C for 20 s and finishing with a final extension at 72 °C for 3 min. The PCR product was purified using 1.0× AMPure XP beads and elution into 12 μL nuclease-free water. The quality and concentration of the linked-tag libraries were analysed on D1000 screentapes and with the Qubit dsDNA HS Assay Kit, respectively.

*Library pooling*. The coverage normalised read-tag library A and B were diluted to 0.9 ng/μL. The linked-tag library was diluted to 0.2 ng/μL. The libraries were pooled by combining 4.6 μL read-tag library A, 4.6 μL read-tag library B and 0.8 μL linked-tag library.

*Sequencing*. The libraries were paired-end (1 × 240 bp and 1 × 25 bp) sequenced on a HiSeq 2500 instrument (Illumina) using onboard clustering and rapid run mode with a HiSeq PE Rapid Cluster Kit v2 (Illumina) and HiSeq Rapid SBS Kit v2, 200 cycles (Illumina). The SBS reagents were supplemented with 9.5 mL Incorporation Master Mix, 9.5 mL Cleavage Reagent Mix and 7 mL Universal Scan Mix to enable sequencing of 265 cycles. The HiSeq was running HiSeq Control Software v.2.2.68 (Illumina) and Real-Time analysis v.1.18.66.3 (Illumina). The libraries were prepared and loaded on the HiSeq using the standard procedures (Illumina: manual #15035786 v.01; manual #15050107 v.02; manual #15061846 v.01) with the following changes. A volume of 10 μL library pool was denatured by adding 10 μL 0.1 N NaOH solution, mixing well by pipetting and incubating for 5 min at 25 °C. The denatured library pool was diluted by adding 980 μL of cold Hybridisation Buffer (Illumina). 400 μL of the denatured and diluted library pool was mixed with 20 μL of denatured and diluted 10 pM PhiX control v3 library (Illumina), and stored on ice until loading. Custom read2 primer mix was prepared by mixing 25 μL of 100 μM f16S_read2_fw and 25 μL of 100 μM f16S_read2_rv in a conical tube (15 mL) and diluted with 4950 μL Hybridisation Buffer (final concentration 0.5 μM). When the paired-end reagent rack was loaded on the HiSeq, the Illumina primer mix in position nr. 16 was replaced with the custom read2 primer mix prepared above. When setting up the HiSeq run in the control software, the standard procedure was followed except for the following steps: For the "Recipe Screen", the following options were chosen: Index type options = No Index. Read 1 cycles = 240. Read 2 cycles = 25. After sequencing, bcl2fastq v2.17.1.14 (Illumina) was used to generate fastq files from bcl files using standard settings (manual #15038058 RevB).

**Short-read amplicon sequencing**. V1–V3 amplicons were made using the 27 F (5′-AGAGTTTGATCCTGGCTCAG-3′)[49] and 534 R (5′-ATTACCGCGGCTGC TGG-3′)[50] primers with barcodes and Illumina adaptors (IDT)[51]. About 25 μL PCR reactions in duplicate were run for each sample using 1X PCRBIO Ultra Mix (PCR Biosystems), 400 nM of both forward and reverse primer and 10 ng template DNA. PCR conditions were 95 °C, for 2 min followed by 20 cycles of 95 °C for 20 s, 56 °C for 30 s and 72 °C for 60 s, followed by a final elongation at 72 °C for 5 min. PCR products were purified using 0.8x AMPure XP beads and eluted in 25 μL nuclease-free water.

V4 amplicons were made using the 515 F (5′-GTGYCAGCMGCCGCGGTAA-3′)[50] and 806 R (5′-GGACTACNVGGGTWTCTAAT-3′)[52] primers. About 25 μL PCR reactions in duplicate were run for each sample using 1X PCRBIO Ultra Mix (PCR Biosystems), 400 nM of both forward and reverse primer and 10 ng template DNA. PCR conditions were 95 °C, for 2 min followed by 30 cycles of 95 °C for 15 s, 55 °C for 15 s and 72 °C for 50 s, followed by a final elongation at 72 °C for 5 min. PCR products were purified using 0.8x AMPure XP beads and eluted in 25 μL nuclease-free water. Two microlitres of purified PCR product from above was used as template for a 25 μL Illumina barcoding PCR reaction containing 1× PCRBIO Reaction buffer, 1 U PCRBIO HiFi Polymerase (PCR Biosystems) and 10 μL of Nextera adaptor mix (Illumina). PCR conditions were 95 °C, for 2 min, 8 cycles of 95 °C for 20 s, 55 °C for 30 s and 72 °C for 60 s, followed by a final elongation at 72 °C for 5 min. PCR products were purified using 0.8x AMPure XP beads and eluted in 25 μL nuclease-free water.

16S rRNA gene V1–V3 and V4 amplicon libraries were pooled separately in equimolar concentrations and diluted to 4 nM. The amplicon libraries were paired-end sequenced (2 × 300 bp) on the Illumina MiSeq using v3 chemistry (Illumina, USA). Ten to 20% PhiX control library was added to mitigate low diversity library effects.

**General bioinformatic methods**. Usearch v.11.0.667[53] was used for processing 16S rRNA gene amplicon data and for read mapping. Mapping of sequences to references was done with the -usearch_global command and the -id 0, -maxaccepts 0, -maxrejects 0, -top_hit_only and -strand plus options unless otherwise stated. UNOISE3[54] was used to resolve exact amplicon sequence variants (ASVs). SINTAX[55] was used for the classification of ASVs using the usearch -sintax command with the -strand plus and -sintax_cutoff 0.8 options. Data were analyzed with R v.4.0.5[56] through RStudio IDE[57], with the tidyverse v.1.3.1 (https://www.tidyverse.org/), vegan v.2.5[58], maps v.3.3.0[59] and Ampvis2 v.2.7.9[60] packages.

**Assembly of full-length 16S rRNA genes**. Raw sequence reads were binned based on the UMIs, and de novo assembled into the synthetic long-read rRNA gene sequences using the fSSU_pipeline_v1.0.sh bash script by Karst et al.[19] available at Github [https://github.com/MadsAlbertsen/fSSU]. The assembled 16S rRNA gene sequences were oriented based on the SILVA 138 SSURef Nr99 database using the usearch v.11.0.667 -orient command and trimmed between the 27f and 1391r primer binding sites using the trimming function in CLC genomics workbench v. 20.0. Sequences without both primer bindings sites were discarded.

**Generation of the MiDAS 4 16S rRNA gene reference database**. The trimmed full-length 16S rRNA genes from above were processed with AutoTax v. 1.5.2[18] to create full-length 16S rRNA gene amplicon sequence variants (FL-ASV) and these were added to the MiDAS 3 reference database[16] to create MiDAS 4.

**Processing of short-read amplicon data**. 16S rRNA gene V1–V3 forward and reverse reads were merged using the usearch -fastq_mergepairs command, filtered to remove phiX sequences using usearch -filter_phix and quality filtered using usearch -fastq_filter with -fastq_maxee 1.0. Dereplication was performed using -fastx_uniques with -sizeout, and amplicon sequence variants (ASVs) were resolved using the usearch -unoise3 command. OTUs clustered at 97% identity were created using the -cluster_otus command. ASV- and OTU-tables were created by mapping the quality-filtered reads to the ASVs using the usearch -otutab command with the -otus or -zotus and -strand plus options. Taxonomy was assigned to OTUs and ASVs using MiDAS 4.8 and the usearch -sintax command with -strand both and -sintax_cutoff 0.8 options.

16S rRNA gene V4 forward reads (reverse reads in relation to the 16S rRNA gene) were trimmed with cutadapt v.2.8[61] based on the V4 primers with the -g ^GGACTACHVGGGTWTCTAAT…TTACCGCGGCKGCTGGCAC and --discard-untrimmed options. The trimmed reads, which span the entire V4 amplicon, were reverse complemented with usearch -fastx_revcomp, and quality filtered using usearch -fastq_filter with -fastq_maxee 1.0. Subsequent processing was like that for the V1–V3 amplicons.

Raw V4 amplicon data from the Global Water Microbiome Consortium project[2] was downloaded from NCBI Sequence Read Archive (SRA) with accession number PRJNA509305 using the SRA-Toolkit v.2.9.2. Forward and reverse reads were merged using the usearch -fastq_mergepairs command, filtered for phiX sequences using usearch -filter_phix, and quality and adaptor filtered using usearch -fastq_filter with the -fastq_stripleft 21, -fastx_stripright 25, and -fastq_maxee 1.0 options. Subsequent processing was like that for the V1–V3 amplicons.

**Construction of phylogenetic trees**. FL-ASVs aligned using SINA v.1.6[62] with the global SILVA 138 NR99 alignment as a reference were obtained from the AutoTax output (temp/FL-ASVs_SILVA_aligned.fa) and loaded into ARB. The multiple alignment was trimmed using the ssuref:bacteria positional variability by parsimony filter to remove highly variable positions. A tree was created from the alignment using RAxML v. 8.2.12 with the raxmlHPC -PTHREADS -m GTRCAT -n score-f -F -p 32323 command and loaded into ARB.

**Microbial community analyses**. Microbial community analyses were performed for samples collected by the MiDAS global consortium. Only activated sludge plants, which include conventional activated sludge (CAS) and sequence batch reactors (SBR), were chosen for detailed analyses. We further selected plants designed for carbon removal (C), carbon removal with nitrification (C,N), carbon removal with nitrification and denitrification (C,N,DN) and carbon removal with nitrogen removal and enhanced biological phosphorus removal, EBPR (C,N,DN,P). Samples with less than 10,000 reads were discarded, providing in total 861 V1–V3 samples and 666 V4 samples.

Associations between the activated sludge microbiota and the following process-related or environmental variables were investigated: process type (as listed above); industrial load (expressed as a fraction of the influent COD); temperature in the process tank (°C), continent and Köppen–Geiger climate classification[63]. Industrial load and temperature was treated as a discrete variable with the following ranges applied: very low (1.8–10.0 °C), low (10.1–15.0 °C), moderate (15.1–20.0 °C), high (20.1–25.0 °C), very high (25.1–30.0 °C), extremely high (30.1–38.0 °C) for temperature; and none (0%), very low (1–10%), low (11–29%), medium (30–50%), high (51–99%), all (100%) for the industrial load. The following climate zone groups were used in the analyses: A: tropical/megathermal climates, B: dry (desert and semi-arid) climates, C: temperate/mesothermal climates, D: continental/microthermal climates, E: polar climates.

For alpha diversity analyses, samples were rarefied to 10,000 reads, and alpha diversity (Observed taxa and inverse Simpsons) was calculated using the ampvis2 package[60]. The Kruskal–Wallis with Dunn's post hoc test (Bonferroni correction with α = 0.01 before correction) was used to determine statistically significant differences in alpha diversity between samples grouped by process and environmental variables.

Distance decay relationship was determined using untransformed values of geographic distance against microbial community similarity distance (Bray–Curtis, or Soerensen) for ASVs, 97% OTUs and genera. Geographical distances between samples were calculated using the distm function in the geosphere R package[64] using the Haversine formula. To examine the strength of correlation between geographic and community distance matrices, the Mantel test using Spearman correlation and 999 permutations was performed using the mantel function in the vegan R package[58].

Beta-diversity distances based on Bray–Curtis (abundance-based) and Soerensen (occurrence-based) for genera (relative genus abundance >0.01% for Soerensen diversity) was calculated using the vegdist function in the vegan R package[58] and visualised by PCoA and RDA plots with the ampvis2 package[60]. Individual process or environmental variables were used as constraints for the RDA. To determine how much individual parameters affected the structure of the microbial community across the WWTPs, a permutational multivariate analysis of variance (PERMANOVA) test was performed on the beta-diversity matrices using the adonis function in the vegan package with 999 permutations.

Core genera and species were defined based on their relative abundances in individual WWTPs. Taxa were defined as abundant when present at >0.1% relative abundance in individual WWTPs. Based on how frequently taxa were observed to be abundant, we defined the following core communities: loose core (>20% of WWTPs), general core (>50% of WWTPs), and strict core (>80% of WWTPs). Additionally, we defined conditionally rare or abundant taxa (CRAT)[30] composed of taxa present in one or more WWTPs at >1% relative abundance, but not belonging to the core taxa.

**Reporting Summary**. Further information on research design is available in the Nature Research Reporting Summary linked to this article.

## Data availability
The raw and assembled sequencing data generated in this study have been deposited in the NCBI SRA database under accession code PRJNA728873. The MiDAS 4 reference database in SINTAX, QIIME and Dada2 format is available at the MiDAS fieldguide website [https://www.midasfieldguide.org/guide/downloads].

## Code availability
R scripts used for data analyses and figures are available at GitHub [https://github.com/msdueholm/MiDAS4][65]. Raw data files for the R scripts are available at Figshare [https://doi.org/10.6084/m9.figshare.16566408.v1][66].

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

## Acknowledgements

The project has been funded by the Danish Research Council (grant 6111-00617 A, P.H.N.) and the Villum Foundation (Dark Matter and grant 13351, P.H.N.). We thank all the involved WWTPs for providing samples and plant metadata. Rune Bakke passed away December 15, 2020. David Jenkins passed away March 6, 2021.

## Author contributions

P.H.N., M.K.D.D., V.R and M.A. designed the study. M.K.D.D., M.A. and P.H.N. wrote the manuscript and all authors reviewed and approved the final manuscript. The MiDAS Global Consortium provided samples and metadata. V.R. and S.K. handled sampling, DNA extraction and library preparation for DNA sequencing. K.S.A. and M.K.D.D. performed the bioinformatics analyses. M.K.D.D., M.N. and K.S.A. carried out statistical analyses.

## Competing interests
The authors declare no competing interests.

## Additional information

## MiDAS Global Consortium

Sonia Arriaga[2], Rune Bakke[3], Nico Boon[4], Faizal Bux[5], Magnus Christensson[6], Adeline Seak May Chua[7], Thomas P. Curtis[8], Eddie Cytryn[9], Leonardo Erijman[10], Claudia Etchebehere[11], Despo Fatta-Kassinos[12], Dominic Frigon[13], Maria Carolina Garcia-Chaves[14], April Z. Gu[15], Harald Horn[16], David Jenkins[17], Norbert Kreuzinger[18], Sheena Kumari[5], Ana Lanham[19], Yingyu Law[20], TorOve Leiknes[21], Eberhard Morgenroth[22], Adam Muszyński[23], Steve Petrovski[24], Maite Pijuan[25], Suraj Babu Pillai[26], Maria A. M. Reis[27], Qi Rong[28], Simona Rossetti[29], Robert Seviour[30], Nick Tooker[31], Pirjo Vainio[32], Mark van Loosdrecht[33], R. Vikraman[34], Jiří Wanner[35], David Weissbrodt[36], Xianghua Wen[37], Tong Zhang[38] & Per H. Nielsen[1]

[2]Environmental Science Department, The Institute for Scientific and Technological Research of San Luis Potosi (IPICYT), San Luis Potosí, Mexico. [3]Department of Process, Energy and Environmental Technology, University College of Southeast Norway, Porsgrunn, Norway. [4]Center for Microbial Ecology and Technology, Ghent University, Ghent, Belgium. [5]Institute for Water and Wastewater Technology, Durban University of Technology, Durban, South Africa. [6]Veolia Water Technologies AB, AnoxKaldnes, Lund, Sweden. [7]Department Of Chemical Engineering, Faculty of Engineering, University of Malaya, Kuala Lumpur, Malaysia. [8]Environmental Engineering, Newcastle University, Newcastle, England. [9]The Cytryn Lab, Microbial Agroecology, Volcani Center, Agricultural Research Organization, Rishon Lezion, Israel. [10]INGEBI-CONICET, University of Buenos Aires, Buenos Aires, Argentina. [11]Department of Biochemistry and Microbial Genetics, Biological Research Institute "Clemente Estable", Montevideo, Uruguay. [12]NIREAS-International Water Research Center, University of Cyprus, Nicosia, Cyprus. [13]Environmental Engineering, McGill University, Montreal, QC, Canada. [14]School of Microbiology, Universidad de Antioquia, Medellín, Colombia. [15]School of Civil and Environmental Engineering, Cornell University, Ithaca, NY, USA. [16]Water Chemistry and Water Technology and DVGW Research Laboratories, Karlsruhe Institute of Technology (KIT), Karlsruhe, Germany. [17]David Jenkins & Associates, Inc, Kensington, CA, USA. [18]Institute for Water Quality and Resource Management, TU Wien, Vienna, Austria. [19]Water Innovation and Research Centre, University of Bath, Bath, England. [20]Singapore Centre of Environmental Life Sciences Engineering (SCELSE) Nanyang Technological University, Singapore, Singapore. [21]Water Desalination and Reuse Center, King Abdullah University of Science and Technology (KAUST), Thuwal, Saudi Arabia. [22]Process Engineering in Urban Water Management, ETH Zürich, Zürich, Switzerland. [23]Department of Biology, Warsaw University of Technology, Warsaw, Poland. [24]Environmental Microbial Genetics Lab, La Trobe University, Melbourne, VIC, Australia. [25]Technologies and Evaluation Area, Catalan Institute for Water Research, ICRA, Girona, Spain. [26]VA Tech Wabag Ltd, Chennai, India. [27]Biochemical Engineering Group, Universidade Nova de Lisboa, Lisboa, Portugal. [28]State Key Laboratory of Environmental Aquatic Chemistry, Research Center for Eco-Environmental Sciences, Chinese Academy of Sciences, Beijing, China. [29]Water Research Institute IRSA - National Research Council (CNR), Rome, Italy. [30]La Trobe University, Melbourne, VIC, Australia. [31]Department of Civil and Environmental Engineering, University of Massachusetts Amherst, Amherst, MA, USA. [32]Kemira Oyj, Espoo R&D Center, Espo, Finland. [33]Environmental Biotechnology, TU Delft, Delft, The Netherlands. [34]VA Tech Wabag, Philippines Inc., Makati City, Philippines. [35]Department of Water Technology and Environmental Engineering, University of Chemistry and Technology, Prague, Czech Republic. [36]Environmental Life Science Engineering, TU Delft, Delft, The Netherlands. [37]School of Environment, Tsinghua University, Beijing, China. [38]Environmental Biotechnology Lab, Department of Civil Engineering, The University of Hong Kong, Hong Kong, Hong Kong.

