## [Peer Review File · Nature Communications]

Reviewers' Comments:

Reviewer #1:

Remarks to the Author:

Summary:

This paper reports on the establishment of a new ecosystem-specific taxonomic database, MiDAS 4.0, which is based on full-length 16S rRNA gene sequences generated through targeted synthetic long read sequencing of DNA extracted from wastewater treatment plants around the world, and curated using the AutoTax pipeline. The authors demonstrate that the new database can provide a higher level of species classification with commonly used primers for short-read amplicon sequencing (e.g. V1-V3, V4). In addition, the new database incorporates apparent novel taxonomic groups, and identifies certain groups of 'core' taxa that are widely abundant in the global WWTP microbiome. This advancement is surely going to be of major use to all researchers studying the wastewater microbiome, and is poised to grow with the generation of high quality metagenome assembled genomes that can be linked to the full length 16S rRNA genes curated here.

While this work represents an exciting breakthrough, there are some minor points that need to be addressed before publication. In particular, the authors should comment on the impact of chimeric sequences in the generation of their curated database. Approximately 0.3% of all sequences represented new phyla or classes. While this is very exciting, some mention of chimera filtering / identification should be made, as this percentage of completely novel and different sequences is well within reported chimera rates for amplicon studies, even with the use of UMIs. As of now, there was no obvious discussion of chimera handling / removal in the methods nor in the paper.

There are additional points, specified below, that should also be addressed, such as the choice of threshold for differential abundance analysis in V1-3 vs. V4 comparisons, and some points on the discussion.

Overall, this paper has great potential to make an immediate impact on the environmental biotechnology field, and is well written and easy to follow.

Specific Comments:

Line 50: This resource recovery could be slightly expanded by listing some examples (e.g. biogas/biomethane, phosphorus, reclaimed water, etc).

Line 56: What 'fundamental level' is referred to here? Genome-level? 16S rRNA gene level?

Line 191: These percentages for ASVs classified at genus level are relative to all ASVs present over 0.01%. Stats were already given for how many of this sequence group could be mapped to MiDAS 4, so why not normalize the genus level classification to what could be mapped? (e.g. it appears that ~100% of mapped sequences to MiDAS4 could be resolved at genus level, whereas ~70% of sequences that mapped to Silva could be resolved at genus level).

Figure 4: The choice of a two-fold difference between the genera seems a bit arbitrary to compare over or underrepresentation, and definitely biases the apparent differences toward the lower abundant taxa (e.g. because a difference of 0.001% could more than double the abundance of a rare taxon). At the very least, this qualification should be discussed.

Line 251: I recommend referring to the reader to Supplementary Figure 2 for this statement. The 'better taxonomic resolution' afforded by V4 is apparent in that figure. However, I do not see any data to indicate lower bias of V4 relative to V1-V3, as there is no comparison to ground truth (e.g. Mock or 'known' composition)

Figure 6: It appears that the relative abundance of the groups (strict core, general core, etc) was determined by 'pooling' all reads from across all global WWTP samples. Another way to calculate those numbers would be on a per sample basis, and report the variance across all applicable WWTPs. In that latter approach, it would reduce the potential that a few samples would highly skew the proportions. Can the authors justify their approach of pooling all reads and taking the proportion of core groups? Was this impacted by any 'outlier' WWTPs?

Figure 7: It should be clarified that these relative abundances are not a mean across process types, but are based on pooling all of the reads from those sample types.

Line 401: It would seem that the more advanced WWTP configurations would select for more specialized bacteria that thrive under the alternating redox configurations employed, but it is not clear whether these organisms are 'versatile' or actually 'specialists' (e.g. PAOs have a pretty defined niche).

Line 405; Any comment on the potential contribution of CRAT to system performance? Or, is it unclear whether they are 'transient' members that are abundant in the influent rather than contributors. Maybe worth citing work (from your group) on mass-balance based approaches to determine their contribution based on growth rates?

Line 469: The abundance of PAOs in C removal plants could also be caused by the presence of a high-F/M selector, which may actually select for large agglomerates (e.g. 'mini granules') and potentially even PAOs if there is redox stratification within such niches. This is indicated by (<https://doi.org/10.1016/j.watres.2020.115865>), although only EBPR plants were investigated.

Line 494: Could the authors discuss whether certain species are truly 'poorly resolved' based on 16S rRNA gene, or rather is this a systematic result of AutoTax species definitions? It seems that having ASVs unclassified at the species level within a genus would indicate that the reference species sequence was not present in the database. Or, can the authors elaborate more on what they mean by this 'poor resolution of species based on the 16S rRNA gene'?

Line 516: This website does not have a doi, and therefore is a bit risky to direct readers here as the website may move domains in 10-20 years from now. Consider moving to a permanent doi, even for this paper.

Reviewer #2:

Remarks to the Author:

This paper presents MiDAS 4 - a 16S rRNA taxonomic database specific for wastewater treatment communities.

I really enjoyed reading this manuscript. The language is fluent and the figures are very nice. I think that development of such a database is critical to the field and has the potential for a wide reach. The authors do a nice job of justifying the merit of the database with sound statistical evidence as well as using the study to glean deeper information on the global make-up of activated sludge communities. I did however have some major comments around potential limitations of the study for overall wastewater treatment systems. Additionally, there are a few instances where I think further interpretation could improve the manuscript (see comments below).

Major Comments:

1. There's almost no mention of archaeal populations or coverage. I think this is a major limitation to the study, especially if MiDAS is to be suitable for all types of wastewater communities – and also regarding the discussion about primer coverage, where V1-V3 and V4 might perform differently for archaea. (And in general I think you should be very careful with how you advise your readers regarding the primers. Really, you have no way of measuring which primer set is showing the "truth" about the community profile.) Do you have any evidence to show how the database performs for archaeal resolution? If not, I think you need to be very clear that this is a CAS database for bacteria, which is still important and useful, but limits the scope for all wastewater treatment technologies.

2. Relatedly, the sampling campaign, while impressive, is biased towards CAS systems – how did the CAS systems compare to the other types of systems included (filters, granule, etc.)?

3. In the section starting on Ln 262 – process and environmental factors... - I assume you only analyzed the samples that came from CAS treatment plants? If you're really only creating an activated sludge database, maybe that should be in the title. Or at least it should be made very clear for the readers. I would imagine that even the granular communities would be different from communities within the conventional systems.

Minor Comments:

4. Section beginning at Ln 153 – the novel taxa – could be discussed in more detail. The results are well-presented, but the discussion and implications feel sparse. At the end of the section I'm left wondering, 'why does that matter?'. And the lines about de novo phyla being artefacts due to the evolutionary rates could be better explained.

5. Ln. 239 Genera should be capitalized.

6. Ln 265 – I assume this is the rarefied richness?

7. Ln 256 – Why did you choose Simpson over, for example, Shannon?

8. Ln 331-333, yeah but that's not really a new idea, is it? Maybe add a reference or two and discuss how your findings compare to what's already been reported.

9. Ln 336 – "frequent", should it be frequently?

10. I think the identification of CRAT genera is very interesting. I wish the manuscript contained more interpretation with respect to them.

11. What is the message in the paragraph beginning on Ln 378? It reads like a catalogue and is missing, perhaps, a final sentence to give meaning to the information presented.

12. Ln 484-485 I think there should be a reference (or two) after this statement.

13. Reference #29 has a funny character.

14. From Supplemental: Were all samples sequenced on both HiSeq and MiSeq platforms?

Reviewer #3:

Remarks to the Author:

Dueholm et al. presented MiDAS 4, a database of full-length 16S rRNA sequences from global wastewater treatment plants (WWTPs). This work significantly expanded from previous efforts of the kind, covering 740 WWTPs from all continents except for Antarctica. The geographical distribution of these plants do represent a global sampling of wastewater as claimed in this manuscript. The study was based on a new sequencing method that generates full-length ASVs

(FL-ASVs), which granted higher taxonomic resolution than classical 16S rRNA amplicons. The authors used SILVA v138 as the reference to annotate the FL-ASVs, and found a significant amount of novelty in sequence and taxonomic diversity. They used AutoTax, recently developed by a largely overlapping group of authors, to curate the taxonomy of FL-ASVs. They identified core and conditionally rare or abundant taxa (CRAT) in these communities and provided detailed discussion on their physiological properties and their functional implications.

In addition, the authors used common primers to sequence V1-V3 and V4 regions of 16S rRNA, and demonstrated that FL-ASV are more effective in taxonomic resolution. This result is largely expected considering the different lengths of the sequences, but it is still valuable to provide the actual benchmarks and notable taxa as a reference to future researchers.

They also established a standard protocol for microbial community analysis of WWTP samples, which I think is valuable.

Overall, I think this is a high-quality study, and the database (MiDAS 4) is a valuable resource to the research community.

However, I do have several concerns, the most important one of which being the lack of more in-depth analyses of the data. Functional analyses based on 16S rRNA data are generally speculative because they do not directly depict the functional genes. In the last paragraph, the authors did mention that they have a MAG dataset that is already published (Singleton et al. Nat Commun. 2021) and it has significant overlap with the biodiversity discovered in the current study. I would recommend that the authors perform an analysis to incorporate those MAG data and explore the correlation between FL-ASV informed functions and the functional genes/pathways discovered in the MAG. This analysis (if supporting the latter) will make the conclusions more compelling.

Another major concern is: Figure 2 shows a phylogenetic tree of the FL-ASVs. By examining this tree, I am a bit concerned with several apparent long branches. For example, a clade colored as Proteobacteria but denoted as new phylum is located at the bottom of the tree. Could these long branches be artifacts in the phylogenetic analysis? The description of this analysis (line 286 of suppl.) is very brief and I cannot judge whether each step has been done correctly. I would encourage the authors to carefully examine their phylogenetics pipeline, and/or consult a phylogenetics specialist, to avoid plausible shortcomings (e.g., less optimal sequence alignment, not removing poorly aligned sequences). This analysis is important especially because it directly impacts the novel biodiversity found in this study.

Figure 5 presents PCoAs using Bray-Curtis and Sørensen which are phylogeny agnostic. I wonder why the authors did not use a phylogenetic metric such as UniFrac, on the phylogenetic tree of FL-ASV which they already reconstructed (Figure 2)? Given that this dataset include much novel biodiversity, the phylogenetic distances among taxa may be highly variable. A phylogenetic metric therefore might provide different insights into the correlation with environmental factors.

The authors should explain what is "MiDAS" (Microbial Database for Activated Sludge) and briefly introduce its background at its first occurrence (line 75). This is particularly important since the name "MiDAS" coincides with the popular microbiome composition analysis tool: MIDAS (Nayfach et al. Genome Res. 2016).

I am not sure if comparing the taxonomic resolution of other databases vs. GTDB r89 is fair, because the latter relies on matching whole-genome sequences. Could the authors justify that? Were there any archaea found in the dataset? The manuscript only mentioned "Ammonia-oxidizing archaea (AOAs) were not detected with MiDAS 4 due to the lack of reference sequences, and because AOAs are not targeted by the V1-V3 primer pair." (line 438). But it does not mention whether the FL-ASV dataset contains any archaeal sequences. It would have been a bit surprising if there is no archaeon living in the wastewater (they even live in the human gut).

Dear Cesar Sanchez,

We appreciate very much that you and the reviewers have taken the time to review our manuscript. We are pleased by the very positive evaluations and the opportunity to resubmit a revised version of our manuscript. We have tried to implement all the constructive suggestions in the revised manuscript. Responses to every point raised in the reviews are provided below in red, italic typeface.

Best regards,

Per Halkjær Nielsen and Morten Kam Dahl Dueholm

Referee #1 (Remarks to the Author):

This paper reports on the establishment of a new ecosystem-specific taxonomic database, MiDAS 4.0, which is based on full-length 16S rRNA gene sequences generated through targeted synthetic long read sequencing of DNA extracted from wastewater treatment plants around the world, and curated using the AutoTax pipeline. The authors demonstrate that the new database can provide a higher level of species classification with commonly used primers for short-read amplicon sequencing (e.g. V1-V3, V4). In addition, the new database incorporates apparent novel taxonomic groups, and identifies certain groups of ‘core’ taxa that are widely abundant in the global WWTP microbiome. This advancement is surely going to be of major use to all researchers studying the wastewater microbiome, and is poised to grow with the generation of high quality metagenome assembled genomes that can be linked to the full length 16S rRNA genes curated here.

We are happy the reviewer can see the major benefits of our database and its applicability for the field. It should be noted that the higher level of classification not only applies at the species level but also for genera.

While this work represents an exciting breakthrough, there are some minor points that need to be addressed before publication. In particular, the authors should comment on the impact of chimeric sequences in the generation of their curated database. Approximately 0.3% of all sequences represented new phyla or classes. While this is very exciting, some mention of chimera filtering / identification should be made, as this percentage of completely novel and different sequences is well within reported chimera rates for amplicon studies, even with the use of UMIs. As of now, there was no obvious discussion of chimera handling / removal in the methods nor in the paper.

The full-length 16S rRNA gene amplicon sequence variants (FL-ASVs) generated in our study are essentially chimera-free. This feature relates back to the use of dual UMI-tagging of the original template molecules before any PCR amplification steps (one unique UMI in each end) which allows filtering of true sequences from chimera already in the synthetic long read assembly (Karst et al. 2018). Analysis of mock community data revealed that a few chimeras were formed despite the strict UMI-filtering, however these were all singleton and were removed in the subsequent amplicon sequence variant (ASV) calling (Dueholm et al. 2020). The lack of chimera and other sequencing errors is one of the unique features of the MiDAS4 database, and this has now been emphasized in the revised manuscript (see line 130-143).

Refs:

Karst, S. M. et al. Retrieval of a million high-quality, full-length microbial 16S and 18S rRNA gene sequences without primer bias. Nature Biotechnology 36, 190–195 (2018).

Dueholm, M. S. et al. Generation of comprehensive ecosystem-specific reference databases with species-level resolution by high-throughput full-length 16S rRNA gene sequencing and automated taxonomy assignment (AutoTax). mBio 11, e01557-20 (2020).

There are additional points, specified below, that should also be addressed, such as the choice of threshold for differential abundance analysis in V1-3 vs. V4 comparisons, and some points on the discussion.

Overall, this paper has great potential to make an immediate impact on the environmental biotechnology field, and is well written and easy to follow.

The choice of threshold for differential abundance is arbitrary and this will be addressed in the revised manuscript (see below). We are glad that the reviewer can see the great potential of the work and find the manuscript well-written.

Specific Comments:

Line 50: This resource recovery could be slightly expanded by listing some examples (e.g. biogas/biomethane, phosphorus, reclaimed water, etc).

We have provided some examples as suggested, see line 54-55

Line 56: What 'fundamental level' is referred to here? Genome-level? 16S rRNA gene level?

It refers to both identity (16S rRNA genes) and general physiology (morphology etc.). We have changed accordingly, see line 58-59.

Line 191: These percentages for ASVs classified at genus level are relative to all ASVs present over 0.01%. Stats were already given for how many of this sequence group could be mapped to MiDAS 4, so why not normalize the genus level classification to what could be mapped? (e.g. it appears that ~100% of mapped sequences to MiDAS4 could be resolved at genus level, whereas ~70% of sequences that mapped to Silva could be resolved at genus level).

Mapping and classification cannot be directly compared. For example, we can have short-read amplicons that map perfectly (100% identity) to several reference sequences with different taxonomies in MiDAS4 and therefore cannot be confidently classified. Similarly, we might be able to assign confident genus-level classification even though the identity is below 99%.

The choice of the 0.01% percent abundance was used to select for taxa which were likely to have a quantitative impact on the ecosystem, while filtering out the rare biosphere which includes many bacteria introduced with the influent wastewaters (Dottorini et al. 2021). This has been emphasized in the revised manuscript, see line 181-183.

Ref.

Dottorini, G. et al. Mass-immigration determines the assembly of activated sludge microbial communities. Proc Natl Acad Sci U S A 118, (2021).

Figure 4: The choice of a two-fold difference between the genera seems a bit arbitrary to compare over or underrepresentation, and definitely biases the apparent differences toward the lower abundant taxa (e.g. because a difference of 0.001% could more than double the abundance of a rare taxon). At the very least, this qualification should be discussed.

The two-fold difference is an arbitrary choice. However, it relates to the uncertainty we usually encounter in amplicon data. The reviewer is correct that the relative abundance of lower abundant taxa is associated with a higher degree of uncertainty, however because we use averaged data from more than 900 samples the

ratios remain stable down to 0.001% relative abundance. This can also be seen in Figure 4a, where the variance seems similar across the relative abundance scale. We have added a small note on this in the revised figure legend manuscript, see line 796-797.

Line 251: I recommend referring to the reader to Supplementary Figure 2 for this statement. The ‘better taxonomic resolution’ afforded by V4 is apparent in that figure. However, I do not see any data to indicate lower bias of V4 relative to V1-V3, as there is no comparison to ground truth (e.g. Mock or ‘known’ composition)

The theoretical taxonomic resolution provided by different variable has previously been determined both in universal systems and in wastewater systems specifically (Johnson et al. 2019; Dueholm et al. 2020). It clearly shows that a much higher taxonomic resolution is provided by V1-V3 than by V4 amplicons. This is, as the reviewer indicates, also clear from the empirical data in Supplementary Figure 2. We will use the above references and Supplementary Figure 2 as support for our claims about higher taxonomic resolution by the V1-V3 primers. In addition, we have toned down the discussion about which primer set is best and emphasized where to use one or the other. We have now also added a comment about archaea, which are not targeted by the V1-V3 primers, but can be identified based on the V4 data. Please see line 238-255.

Refs.

Johnson, J. S. et al. Evaluation of 16S rRNA gene sequencing for species and strain-level microbiome analysis. Nature Communications 10, 1–11 (2019).

Dueholm, M. S. et al. Generation of comprehensive ecosystem-specific reference databases with species-level resolution by high-throughput full-length 16S rRNA gene sequencing and automated taxonomy assignment (AutoTax). mBio 11, e01557-20 (2020).

Figure 6: It appears that the relative abundance of the groups (strict core, general core, etc) was determined by ‘pooling’ all reads from across all global WWTP samples. Another way to calculate those numbers would be on a per sample basis, and report the variance across all applicable WWTPs. In that latter approach, it would reduce the potential that a few samples would highly skew the proportions. Can the authors justify their approach of pooling all reads and taking the proportion of core groups? Was this impacted by any ‘outlier’ WWTPs?

Thanks for the comment. The relative abundance of the groups was determined based on relative abundances for ASV in each sample, which means that each sample is treated equally. Because we use combined data from more than 900 samples it is unlikely that outliers will have any detectable impact on the results. The choice to summarize the data was to make the figure easier to understand and the overall conclusion stand out clearly. This has been included in the revised figure legend, please see line 819-821.

Figure 7: It should be clarified that these relative abundances are not a mean across process types, but are based on pooling all of the reads from those sample types.

The read counts for each sample were normalized (converted to relative abundances) before calculating the mean relative abundance for each process type. Therefore, we believe that our statement is correct. We have emphasized this in the revised manuscript, please see line 819-821.

Line 401: It would seem that the more advanced WWTP configurations would select for more specialized bacteria that thrive under the alternating redox configurations employed, but it is not clear whether these organisms are ‘versatile’ or actually ‘specialists’ (e.g. PAOs have a pretty defined niche).

This is an interesting question. Usually a versatile organism (generalist) is defined by thriving under several environmental conditions and by consuming many different substrates. A specialist can only use few

substrates and grow under few conditions (Székely and Langenheder 2014). In our case, the advanced plants provide a range of redox conditions and also many substrates, and the bacteria enriched here can grow under many different conditions. They can also grow in the simple C-removing plants only, but may be outcompeted by specialist. Thus, we see them as more versatile than those growing only in the simple C-removing plants that cannot use different electron acceptors, usually consume only few substrates, and do not have the poly-P metabolism. So, we regard the bacteria enriched in the advanced plants as being versatile and not specialist in terms of physiology. It is clear that they may be regarded as PAO specialists, but not general specialists. So, we have kept the wording.

Ref.

Székely, A. J. & Langenheder, S. The importance of species sorting differs between habitat generalists and specialists in bacterial communities. FEMS Microbiology Ecology 87, 102–112 (2014).

Line 405; Any comment on the potential contribution of CRAT to system performance? Or, is it unclear whether they are ‘transient’ members that are abundant in the influent rather than contributors. Maybe worth citing work (from your group) on mass-balance based approaches to determine their contribution based on growth rates?

The global MiDAS survey does not include time series or influent data, therefore, we cannot say anything conclusive about the general implications of CRAT taxa on system performance. However, as the reviewer suggests, we have added a few words about this and clarify that they can be present due to short-term mass immigration (Dottorini et al. 2021) or specific operational conditions, and in both cases potentially affect the plant operation. This has been included, please see line 368-374.

Ref.

Dottorini, G. et al. Mass-immigration determines the assembly of activated sludge microbial communities. Proc Natl Acad Sci U S A 118, (2021).

Line 469: The abundance of PAOs in C removal plants could also be caused by the presence of a high-F/M selector, which may actually select for large agglomerates (e.g. ‘mini granules’) and potentially even PAOs if there is redox stratification within such niches. This is indicated by (<https://doi.org/10.1016/j.watres.2020.115865>), although only EBPR plants were investigated.

Thanks for the comment. This could also be the case. However, we do not have any plant data on the F/M-ratio and SVI to support such a claim. The presence of GAOs and PAOs in simple plants has been observed before (Mao et al. 2015) and it may relate to their capability to uptake and store large amount of substrate under dynamic conditions, e.g. with high F/M selectors, also under aerobic conditions. However, as we do not have data to support this we have not included more in the revised manuscript.

Ref.

Mao, Y., Graham, D. W., Tamaki, H. & Zhang, T. Dominant and novel clades of Candidatus Accumulibacter phosphatis in 18 globally distributed full-scale wastewater treatment plants. Sci Rep 5, 11857 (2015).

Line 494: Could the authors discuss whether certain species are truly ‘poorly resolved’ based on 16S rRNA gene, or rather is this a systematic result of AutoTax species definitions? It seems that having ASVs unclassified at the species level within a genus would indicate that the reference species sequence was not present in the database. Or, can the authors elaborate more on what they mean by this ‘poor resolution of species based on the 16S rRNA gene’?

There can be different reasons that a 16S rRNA gene cannot be classified at the species-level despite having a perfect match in the reference database. 1) Different species may have identical full-length 16S rRNA

genes. The textbook example is Escherichia and Shigella. 2) Some bacteria have multiple copies of the 16S rRNA gene which are more diverse than homologs in another species. 3) The variable regions (e.g., V4) can be conserved despite variation in the full-length 16S rRNA genes. We have updated the manuscript to reflect this, see line 439-442

Line 516: This website does not have a doi, and therefore is a bit risky to direct readers here as the website may move domains in 10-20 years from now. Consider moving to a permanent doi, even for this paper.

We agree that this could be a potential future problem. However, we consider the risk low, and as several published articles already refers to the website, changing the current URL to a persistent URL may provide more problems than gains.

Reviewer #2 (Remarks to the Author):

This paper presents MiDAS 4 - a 16S rRNA taxonomic database specific for wastewater treatment communities.

I really enjoyed reading this manuscript. The language is fluent and the figures are very nice. I think that development of such a database is critical to the field and has the potential for a wide reach. The authors do a nice job of justifying the merit of the database with sound statistical evidence as well as using the study to glean deeper information on the global make-up of activated sludge communities. I did however have some major comments around potential limitations of the study for overall wastewater treatment systems. Additionally, there are a few instances where I think further interpretation could improve the manuscript (see comments below).

We are glad to hear that the reviewer finds the manuscript important and well-written.

Major Comments:

1. There's almost no mention of archaeal populations or coverage. I think this is a major limitation to the study, especially if MiDAS is to be suitable for all types of wastewater communities – and also regarding the discussion about primer coverage, where V1-V3 and V4 might perform differently for archaea. (And in general I think you should be very careful with how you advise your readers regarding the primers. Really, you have no way of measuring which primer set is showing the “truth” about the community profile.) Do you have any evidence to show how the database performs for archaeal resolution? If not, I think you need to be very clear that this is a CAS database for bacteria, which is still important and useful, but limits the scope for all wastewater treatment technologies.

Thanks for this constructive comment. The reviewer is correct that we have focus almost exclusively on bacteria in the current study. The reason for this is that our primer-based approach for full-length 16S rRNA gene sequencing unfortunately does not target archaea. The only archaeal references in MiDAS 4 database are therefore those which were added when we created MiDAS 3 (for details see Nierychlo et al. 2020). The V1-V3 primers, which were used for most analyses, do not target archaea either. Accordingly, we did not notice that we were missing any archaea. To estimate the relative abundance of archaea we have now re-analyzed their relative abundance across all WWTP based on the V4 amplicon dataset. We can see that the relative abundance of archaea is generally low (median = 0.18%, see figure below). However, for a few WWTPs it was very high (up to 11.7%) and they should therefore not be neglected. To address archaea properly in the revised manuscript we have added the above information in the amplicon primer discussion. In addition, we have toned down the discussion about which primer set is best and emphasized where to use one or the other. See line 200-203 and 238-255.

Figure caption: Relative read abundance of archaeal ASV in the MiDAS global V4 amplicon dataset classified with SILVA 138 SSURef NR99 for activated sludge samples belonging to the fore main process types (C: carbon removal; C,N: carbon removal with nitrification; C,N,DN: carbon removal with nitrification and denitrification; C,N,DN,P: carbon removal with nitrogen removal and enhanced biological phosphorus removal).

Ref.:

Nierychlo, M. et al. MiDAS 3: An ecosystem-specific reference database, taxonomy and knowledge platform for activated sludge and anaerobic digesters reveals species-level microbiome composition of activated sludge. *Water Research* 182, 115955 (2020).

2. Relatedly, the sampling campaign, while impressive, is biased towards CAS systems – how did the CAS systems compare to the other types of systems included (filters, granule, etc.)?

The WWTPs included in the current study were selected by the volunteering sampling coordinators for each country and their access to WWTPs. Because CAS is the most common WWTP design, the sampling was biased towards this plant type. This is also the reason that we focus mainly on the CAS plants in our paper. However, the MiDAS 4 database performs very well on other plant types too. See figure below which is based on V1-V3 amplicon data from the current study. The figure will be included as Supplementary Figure 2b in the revised manuscript.

3. In the section starting on ln 262 – process and environmental factors... - I assume you only analyzed the samples that came from CAS treatment plants? If you're really only creating an activated sludge database, maybe that should be in the title. Or at least it should be made very clear for the readers. I would imagine that even the granular communities would be different from communities within the conventional systems.

As shown above, we have created a reference database for all types of WWTPs. The fact the database includes high identity references for almost the same percentage of ASVs independent of plant type, suggest that many taxa are shared across plant types. When we look more detailed into our results, we do also see that the same species are present in activated and biofilm and granular sludge systems– they are just often present in other relative abundances. So, our data do not indicate any major difference in the microbial communities across different reactor types as long as the overall processes are similar. This is now included in the manuscript, see line 187-191.

Minor Comments:

4. Section beginning at ln 153 – the novel taxa – could be discussed in more detail. The results are well-presented, but the discussion and implications feel sparse. At the end of the section I'm left wondering, 'why does that matter?'. And the lines about de novo phyla being artefacts due to the evolutionary rates could be better explained.

We agree that the implications of the high number of de novo taxa lack a concluding statement and have in the revised manuscripts addressed this, see line 152-154. We have also tried to clarify the discussion about the novel phyla being artifacts, see line 156-167.

5. Ln. 239 Genera should be capitalized.

We have corrected this.

6. Ln 265 – I assume this is the rarefied richness?

This is correct, and we have emphasized this in the revised manuscript, see line 258.

7. Ln 256 – Why did you choose Simpson over, for example, Shannon?

The specific diversity index is a matter of taste, and they generally show the same trend. The inverse Simpsons index provides a more intuitive meaning to us. It is defined as the effective number of types that is obtained when the weighted arithmetic mean is used to quantify average proportional abundance of types in the dataset of interest.

8. Ln 331-333, yeah but that's not really a new idea, is it? Maybe add a reference or two and discuss how your findings compare to what's already been reported.

Using taxonomy (genus and species classifications) to identify core taxa has to our knowledge not been done before, and it is only possible because of the high genus and species-level classification rates achieved using MiDAS 4. Previous studies have identified core taxa based on OTUs of which many were poorly classified (family level or above). It is therefore not possible to compare those data with ours.

9. Ln 336 – “frequent”, should it be frequently?

It should be frequently, and this has been corrected in the revised manuscript.

10. I think the identification of CRAT genera is very interesting. I wish the manuscript contained more interpretation with respect to them.

We agree that these taxa are very interesting. Unfortunately, it is not possible to provide clear interpretation about their biological roles from the current data set, as we only have two samples from each WWTP and therefore lack the temporal component, which is important to study these taxa. We are using our longitudinal data from Danish WWTP in combination with single cell in situ techniques to study individual CRAT taxa to learn more about these. For an example see our recent work on Microthrix, which is related to sludge bulking (Nierychlo et al. 2021). We have addressed this in the revised manuscript, see line 368-374.

Ref.

Nierychlo, M. et al. Low Global Diversity of Candidatus Microthrix, a Troublesome Filamentous Organism in Full-Scale WWTPs. Front Microbiol 12, 690251 (2021).

11. What is the message in the paragraph beginning on Ln 378? It reads like a catalogue and is missing, perhaps, a final sentence to give meaning to the information presented.

Thank for the comment. The list of core and CRAT species can be used as a guide to select which species should gain priority for further studies. We have added a final sentence about this in the revised manuscript, see line 368-374.

12. Ln 484-485 I think there should be a reference (or two) after this statement.

We are not able to see where the above comment fits in. The provided line numbers refer to figure 9.

13. Reference #29 has a funny character.

“Astudillo García” should have been “Astudillo-García“. This has been corrected in the revised manuscript.

14. From Supplemental: Were all samples sequenced on both HiSeq and MiSeq platforms?

No, synthetic long read full-length 16S rRNA were only sequenced on the HiSeq. This has been corrected in table SI-1

Reviewer #3 (Remarks to the Author):

Dueholm et al. presented MiDAS 4, a database of full-length 16S rRNA sequences from global wastewater treatment plants (WWTPs). This work significantly expanded from previous efforts of the kind, covering 740 WWTPs from all continents except for Antarctica. The geographical distribution of these plants do represent a global sampling of wastewater as claimed in this manuscript. The study was based on a new sequencing method that generates full-length ASVs (FL-ASVs), which granted higher taxonomic resolution than classical 16S rRNA amplicons. The authors used SILVA v138 as the reference to annotate the FL-ASVs, and found a significant amount of novelty in sequence and taxonomic diversity. They used AutoTax, recently developed by a largely overlapping group of authors, to curate the taxonomy of FL-ASVs. They identified core and conditionally rare or abundant taxa (CRAT) in these communities and provided detailed discussion on their physiological properties and their functional implications.

In addition, the authors used common primers to sequence V1-V3 and V4 regions of 16S rRNA, and demonstrated that FL-ASV are more effective in taxonomic resolution. This result is largely expected considering the different lengths of the sequences, but it is still valuable to provide the actual benchmarks and notable taxa as a reference to future researchers.

They also established a standard protocol for microbial community analysis of WWTP samples, which I think is valuable.

Overall, I think this is a high-quality study, and the database (MiDAS 4) is a valuable resource to the research community.

Thanks for the comments. We are happy that the reviewer values the extent of our survey and the importance of the MiDAS 4 database for the research community.

However, I do have several concerns, the most important one of which being the lack of more in-depth analyses of the data. Functional analyses based on 16S rRNA data are generally speculative because they do not directly depict the functional genes. In the last paragraph, the authors did mention that they have a MAG dataset that is already published (Singleton et al. Nat Commun. 2021) and it has significant overlap with the biodiversity discovered in the current study. I would recommend that the authors perform an analysis to incorporate those MAG data and explore the correlation between FL-ASV informed functions and the functional genes/pathways discovered in the MAG. This analysis (if supporting the latter) will make the conclusions more compelling.

*We agree that it would be great to combine the 16S rRNA study with the MAG study. However, that is outside the scope of this paper and would be very comprehensive and not something that could be included in our present paper if we should go in any details. Instead, we are taking another approach: we are studying various specific groups in more details in specific studies, where we combine the 16S rRNA data with the MAG data and a comprehensive phylogenetic analysis plus FISH probe design and in situ analysis of the ecophysiology by FISH-Raman microspectroscopy. We have done that already for the filamentous bacteria *Candidatus Microthrix* (Nierychlo et al 2021) and several more groups are underway. We focus on groups that are of special interest and each of them is a full paper. The present paper presents the fundamental full-length global reference database and the detailed analysis of this, which we find it in its own is a full paper worth as it provides the foundation for further studies in the field.*

Another point is that we are still missing high-quality MAGs from WWTPs globally, and it is unclear whether functional genes or pathways are conserved across genera and species in different parts of the world. The link between FL-ASVs and functional genes/pathways will therefore be associated with a high degree of uncertainty.

Ref.

*Nierychlo, M. et al. Low Global Diversity of *Candidatus Microthrix*, a Troublesome Filamentous Organism in Full-Scale WWTPs. *Front Microbiol* 12, 690251 (2021).*

Another major concern is: Figure 2 shows a phylogenetic tree of the FL-ASVs. By examining this tree, I am a bit concerned with several apparent long branches. For example, a clade colored as Proteobacteria but denoted as new phylum is located at the bottom of the tree. Could these long branches be artifacts in the phylogenetic analysis? The description of this analysis (line 286 of suppl.) is very brief and I cannot judge whether each step has been done correctly. I would encourage the authors to carefully examine their phylogenetics pipeline, and/or consult a phylogenetics specialist, to avoid plausible shortcomings (e.g., less optimal sequence alignment, not removing poorly aligned sequences). This analysis is important especially because it directly impacts the novel biodiversity found in this study.

Thanks for the comment. It should be noted that we do not use phylogeny to define novel taxa. This is naively done using the percent identity threshold provided by Yarza et al. 2014. The de novo placeholder names are a useful tool, but they should not be viewed as a replacement for phylogenetic analyses. The great benefit of the placeholder names is that they can be used to pinpoint lineages, which are of importance for the ecosystem and that they are easy to recognize across studies. These can subsequently be examined via targeted genome-based phylogenetic analyses as described above. The tree in Figure 2 is used as a simple

depiction of the diversity in the ecosystems, and the branch lengths are of minor importance. We have improved the discussion in the revised manuscript, see line 156-167.

Figure 5 presents PCoAs using Bray-Curtis and Sørensen which are phylogeny agnostic. I wonder why the authors did not use a phylogenetic metric such as UniFrac, on the phylogenetic tree of FL-ASV which they already reconstructed (Figure 2)? Given that this dataset include much novel biodiversity, the phylogenetic distances among taxa may be highly variable. A phylogenetic metric therefore might provide different insights into the correlation with environmental factors.

The rationale behind using UniFrac would be that important traits scales linearly with the phylogenetic distance between ASVs. However, we know that many of the important traits are categorical (yes/no) and only conserved at lower taxonomic ranks (genus/species). We found a high classification rate at the genus-level and since many genera are shared on a global scale, this provides a unique possibility to confidently determine the genus-level taxonomic diversity, so we believe that this data is more relevant. A discussion about our choice have been added in the revised manuscript, see line 283-288.

The authors should explain what is “MiDAS” (Microbial Database for Activated Sludge) and briefly introduce its background at its first occurrence (line 75). This is particularly important since the name “MiDAS” coincides with the popular microbiome composition analysis tool: MIDAS (Nayfach et al. Genome Res. 2016).

We have expanded our introduction to MiDAS to avoid any confusion with the microbiome composition analysis tool, see line 72-80.

I am not sure if comparing the taxonomic resolution of other databases vs. GTDB r89 is fair, because the latter relies on matching whole-genome sequences. Could the authors justify that?

We think this comment relies on a misunderstanding. We do not use the whole-genome database in GTDB but the GTDB r89 16S rRNA gene reference database, which includes all 16S rRNA genes found in the GTDB r89 MAG database. We have highlighted this in the revised manuscript to avoid confusion, see line 206-209.

Were there any archaea found in the dataset? The manuscript only mentioned “Ammonia-oxidizing archaea (AOAs) were not detected with MiDAS 4 due to the lack of reference sequences, and because AOAs are not targeted by the V1-V3 primer pair.” (line 438). But it does not mention whether the FL-ASV dataset contains any archaeal sequences. It would have been a bit surprising if there is no archaeon living in the wastewater (they even live in the human gut).

This is an important point, and also raised by reviewer #2. The only archaeal references in the MiDAS 4 reference database are those which were added when we created MiDAS 3 (for details see Nierychlo et al. 2020). In the present study no new archaeal FL-ASVs were retrieved for the reference database because the primers used did not target archaea. In the amplicon study we used both the V1-V3 primers and the V4 primers where only the latter target some (but not all) archaea. To estimate the relative abundance of archaea in the WWTPS worldwide we have now re-analyzed their relative abundance across all WWTP based on the V4 amplicon dataset. We can see that the relative abundance of archaea is generally low (median = 0.18%, see figure below). However, for a few WWTPs it was high (up to 11.7%) and they should therefore not be neglected. To address the archaea properly in the revised manuscript we have added the above information in the amplicon primer discussion. In addition, we have toned down the discussion about which primer set is best and emphasized where to use one or the other. See line 200-203 and 238-255.

Figure caption: Relative read abundance of archaeal ASV in the MiDAS global V4 amplicon dataset classified with SILVA 138 SSURef NR99 for activated sludge samples belonging to the four main process types (C: carbon removal; C,N: carbon removal with nitrification; C,N,DN: carbon removal with nitrification and denitrification; C,N,DN,P: carbon removal with nitrogen removal and enhanced biological phosphorus removal).

Ref.

Nierychlo, M. et al. MiDAS 3: An ecosystem-specific reference database, taxonomy and knowledge platform for activated sludge and anaerobic digesters reveals species-level microbiome composition of activated sludge. *Water Research* 182, 115955 (2020).

Reviewers' Comments:

Reviewer #1:

Remarks to the Author:

The authors have adequately addressed all of my comments in this revision. I appreciate their thorough responses.

Reviewer #2:

Remarks to the Author:

The authors thoroughly addressed all the comments raised in the first review. I think the manuscript is well-revised and marks a substantial contribution to the field.

Reviewer #3:

Remarks to the Author:

The authors have made decent efforts in addressing the three reviewers' comments. I appreciate that the authors have extended the discussion of the lack of archaeal signals in the dataset. I accept the authors' justification for not including the MAG analysis. I am not sure if the authors' argument for not using UniFrac is valid, because UniFrac models the evolutionary distances among community members, not their traits. But I think this question is less important and shouldn't block the publication of the result (Fig. 5) which delivers a clear message using commonly used metrics.

Finally and however, I am not fully convinced by the authors' justification for not examining the phylogenetic pipeline. I see that the taxonomic groups were determined by percent identity thresholds. However, I am concerned that the several unusually long branches happen to be novel phyla (which is one of the important conclusions of this entire paper). Could it be -- I am suspecting -- that the percent identity AND phylogenetic distance were both overestimated because the sequence alignment was less accurate? For example, some non-homologous sequences in the intergenic or variable regions of the rRNA sequences were incorrectly aligned? We know that the alignment of rRNA sequences is a difficult bioinformatic task due to these concerns. To check this, the authors may consider pruning low-variance regions off the alignment and re-assess the percent identity values to see if they are strongly correlated with the original evaluations.

Overall, I think (and agree with other reviewers) that this is a high-quality manuscript and its publication will be impactful to the field. I hope that the authors can resolve my last concern, which might also be a concern from the potential audience when they read the figure.

Dear Cesar Sanchez,

We appreciate very much that you and the reviewers have taken the time to review our manuscript a second time. We are pleased to hear that two of the reviewers recommend acceptance of the article as it is, and that the last one only has minor comments. We have addressed these comments here and in the revised manuscript. Responses to every point raised are provided below in red, italic typeface. In addition, we have addressed the editorial requests in the manuscript, reporting summary, and author checklist. We also like to include the first author, Morten Dueholm as co-corresponding author in addition to Per H. Nielsen.

Best regards,

Per Halkjær Nielsen and Morten Kam Dahl Dueholm

Reviewer #1 (Remarks to the Author):

The authors have adequately addressed all of my comments in this revision. I appreciate their thorough responses.

We are happy to hear that the reviewer is satisfied with our responses and changes in the manuscript.

Reviewer #2 (Remarks to the Author):

The authors thoroughly addressed all the comments raised in the first review. I think the manuscript is well-revised and marks a substantial contribution to the field.

We are glad that the reviewer is satisfied with how we have addressed the comments raised in the first review.

Reviewer #3 (Remarks to the Author):

The authors have made decent efforts in addressing the three reviewers' comments. I appreciate that the authors have extended the discussion of the lack of archaeal signals in the dataset. I accept the authors' justification for not including the MAG analysis. I am not sure if the authors' argument for not using UniFrac is valid, because UniFrac models the evolutionary distances among community members, not their traits. But I think this question is less important and shouldn't block the publication of the result (Fig. 5) which delivers a clear message using commonly used metrics.

We appreciate that the reviewer is satisfied with how we have addressed most comments raised in the first review.

Finally and however, I am not fully convinced by the authors' justification for not examining the phylogenetic pipeline. I see that the taxonomic groups were determined by percent identity thresholds. However, I am concerned that the several unusually long branches happen to be novel phyla (which is one of the important conclusions of this entire paper). Could it be -- I am suspecting -- that the percent identity AND phylogenetic distance were both overestimated because the sequence alignment was less accurate? For example, some non-homologous sequences in the intergenic or variable regions of the rRNA sequences were incorrectly aligned? We know that the alignment of rRNA sequences is a difficult bioinformatic task due to these concerns. To check this, the authors may consider pruning low-variance regions off the alignment and re-assess the percent identity values to see if they are strongly correlated with the original evaluations.

We appreciate the comments and acknowledge that it is not stated clear enough how we have determined the percent identity and created the trees. We have elaborated on this below:

Identification of new putative taxa: The percent identity used to classify FL-ASVs as de novo taxa does not originate from a multiple alignment but from pairwise alignments created between each FL-ASVs and their closest relatives in the SILVA database based on the usearch algorithm. This has now been clarified in the manuscript (l. 1133). We have observed that this approach is resulting in higher percent identity (i.e., better alignments) compared to those obtained from multiple alignments.

Construction of phylogenetic trees: Regarding the long branches on the phylogenetic tree, the reviewer is correct that they could be a result of an imperfect alignment. However, manual curation of a nearly 100,000 sequence multiple alignment is not practically possible, and due to reproducibility, it is not possible to selectively curate alignments manually for sequences that create unusual long branches.

To determine if the long branches were due to poor alignments, we did a manual inspection of the alignment for selected FL-ASVs representing de novo phyla. However, we did not observe any issues with the alignments for those sequences. The raw SINA multiple alignment has been made available for the reviewers through the following Dropbox-link:

https://www.dropbox.com/s/cvsilsv56u2vw74/FLASVs_ssufef_bac_for_tree.fasta?dl=0

It should be noted that the alignment used to create the trees in Figure 2 was pruned and only contain those positions that are conserved according to the *ssufef:bacteria* positional variability by parsimony filter in ARB. This information has now been added to the figure legend (l. 1127-1130 and l. 803-806). To investigate how the choice of pruning filter impacts the tree, we created four new trees with FastTree2 using the alignment used for the trees in figure 2 and base frequencies filtered alignments (30%, 50%, and 70% base conservation). The result is shown below and reveals that the long branches are present in all trees. This indicates that the long branches are caused by non-homologous sequences in the intergenic or variable regions and likely represent true diversification.

Figure R1: Phylogenetic trees created using FastTree v. 2.1.11 with the -nt, -gtr, and -gamma options based on the SINA multiple alignment from the paper pruned using the *ssu-ref:bacteria* positional variability by parsimony filter or custom base frequency filters (30%, 50%, and 70% conservation). The trees are colored according to sequence novelty. Sequence novelty was determined by mapping the FL-ASVs against the SILVA 138 SSURef Nr99 database with Usearch v.11.0.667 with the -usearch_global command and the -id 0, -maxrejects 0, -maxrejects 0, and -top_hits_only options. Novelty was defined as follows: New species <98.7% identity; New genus <94.5% identity; New family <86.5% identity; New order <82.0% identity; New class <78.5% identity; New phylum <75.0% identity.

Finally, we would like to emphasize that in our mind the identification of new potential phyla is a less important part of the paper, because these taxa are in very low abundance in the ecosystem. Accordingly, they are assumed to have no or little effect on the wastewater treatment process.

Overall, I think (and agree with other reviewers) that this is a high-quality manuscript and its publication will be impactful to the field. I hope that the authors can resolve my last concern, which might also be a concern from the potential audience when they read the figure.

We are delighted that the reviewer recognizes the quality and importance of the paper, and we hope that our comments above will address the reviewer's remaining concerns.